# The gut commensal *Blautia* maintains colonic mucus function under low-fiber consumption through secretion of short-chain fatty acids

Sandra M. Holmberg [1,2,7], Rachel H. Feeney [1,2,7], Vishnu Prasoodanan P.K.[1,2,7], Fabiola Puértolas-Balint [1,2], Dhirendra K. Singh[1,2], Supapit Wongkuna [1,2], Lotte Zandbergen[1,2], Hans Hauner[3,4], Beate Brandl[5], Anni I. Nieminen [6], Thomas Skurk [5] & Bjoern O. Schroeder [1,2] ✉

Beneficial gut bacteria are indispensable for developing colonic mucus and fully establishing its protective function against intestinal microorganisms. Low-fiber diet consumption alters the gut bacterial configuration and disturbs this microbe-mucus interaction, but the specific bacteria and microbial metabolites responsible for maintaining mucus function remain poorly understood. By using human-to-mouse microbiota transplantation and ex vivo analysis of colonic mucus function, we here show as a proof-of-concept that individuals who increase their daily dietary fiber intake can improve the capacity of their gut microbiota to prevent diet-mediated mucus defects. Mucus growth, a critical feature of intact colonic mucus, correlated with the abundance of the gut commensal *Blautia*, and supplementation of *Blautia coccoides* to mice confirmed its mucus-stimulating capacity. Mechanistically, *B. coccoides* stimulated mucus growth through the production of the short-chain fatty acids propionate and acetate via activation of the short-chain fatty acid receptor Ffar2, which could serve as a new target to restore mucus growth during mucus-associated lifestyle diseases.

Colonic mucus forms a crucial barrier between the intestinal epithelium and the great number of gut bacteria. During homeostasis, the most inner part of the mucus, which is adjacent to the epithelium ("inner layer"), is virtually sterile, while the more luminal part of this highly glycosylated gel network is intermixed with bacteria and intestinal content and is thus less defined[1]. In addition to being a physical barrier, mucus has intrinsic expansion activity, likely due to a combination of continual baseline mucus secretion and swelling. This results in a constant luminal-directed flow toward the gut bacteria, thereby actively pushing microorganisms away from the host epithelium. This "mucus growth" is measured to be ~2 μm/min in mice and ~4 μm/min in humans[2,3].

While initially only being regarded as a lubricant for fecal material, accumulating evidence suggests that a compromised mucus layer is associated with several human diseases. For example, bacterial penetration and colonization of the mucus layer has been observed in

[1]Department of Molecular Biology, Umeå University, Umeå, Sweden. [2]Laboratory for Molecular Infection Medicine Sweden (MIMS) and Umeå Center for Microbial Research (UCMR), Umeå University, Umeå, Sweden. [3]Institute in Nutritional Medicine, TU Munich, Munich, Germany. [4]TU Munich, School of Medicine, Munich, Germany. [5]ZIEL Institute for Food and Health, TU Munich, Munich, Germany. [6]Institute for Molecular Medicine Finland (FIMM), University of Helsinki, Helsinki, Finland. [7]These authors contributed equally: Sandra M. Holmberg, Rachel H. Feeney, Vishnu Prasoodanan P.K. ✉e-mail: bjorn.schroder@umu.se

metabolic disease[4] and in inflammatory bowel diseases (IBDs)[5–8]. In patients with ulcerative colitis, one of the two major forms of IBD, mucus defects even precede the onset of disease and may thus be an important contributing factor[8]. Likewise, mice deficient in mucin-2 (Muc2), the structural scaffold protein forming the colonic mucus, or mice lacking the predominant O-linked oligosaccharides that decorate Muc2, develop severe colitis[7,9–11], thus causally linking mucus dysfunction to intestinal inflammation.

Recent studies have identified the gut microbiota as an indispensable factor for proper mucus function. Germ-free mice have a penetrable mucus layer that requires microbial colonization over at least 6 weeks to become impenetrable[12]. In addition, a mucus penetrability phenotype can be transferred to germ-free mice via a microbiota transplant[13] or between mice through cohousing[14], further substantiating that the microbiota can modulate mucus function.

The intestinal microbiota configuration is strongly dependent on diet[15] and specifically on dietary fiber[16,17]. However, modern dietary habits in the industrialized world are often characterized by highly processed foods that are rich in simple sugars and saturated fatty acids but contain low amounts of plant-derived dietary fiber. Consumption of such a low-fiber "Western-style diet" (WSD) in mice not only leads to a reduction in bacterial diversity and extinction of important bacterial taxa over generations[18] but is also associated with the development of IBDs[19,20]. Accordingly, upon feeding of a low-fiber diet, the bacterial community in gnotobiotic and specific-pathogen-free mice can degrade the protective colonic mucus layer, thereby triggering inflammation and increasing susceptibility to intestinal infection[21–23].

While microbial enzymatic mucus degradation is known to compromise the integrity of the mucus layer[21,24–27], limited knowledge exists about how gut bacteria promote mucus function. Microbial metabolites, including indoleacrylic acid[28] or butyrate[29], have been identified to stimulate Muc2 mRNA expression, but Muc2 expression does not necessarily correlate with mucus function, including mucus thickness, penetrability and growth, which is largely regulated on a post-translational level. However, by using an ex vivo method to investigate mucus function on viable tissue[2], Escherichia coli-derived lipopolysaccharide (LPS), its subcomponent lipid A, and flagellin from Bacillus subtilis were all identified to stimulate mucus secretion by a distinct sentinel goblet cell located at the crypt opening[3]. Although this process, termed compound exocytosis, is able to clear invading bacteria from the crypt opening, this mechanism appeared nonessential for mucus maintenance under homeostatic conditions and is instead an acute response following mucus barrier penetration.

In previous work, we showed that microbiota transplantation from mice fed a high-fiber diet to mice fed a low-fiber WSD could prevent defects in mucus growth rate and mucus penetrability[30], supporting the crucial influence of a defined commensal gut microbiota configuration on mucus function[31]. Motivated by the decreasing dietary fiber consumption in industrialized societies and the associated detrimental effect on mucus function, we aimed to investigate whether increased dietary fiber intake improves the microbiota-mucus interaction for human microbiota. Here we show, by using human-to-mouse microbiota transplantation, metabolomics and an ex vivo explant method to study mucus function on viable tissue, that increased fibre intake in humans enriches for the beneficial gut bacterium Blautia, and that B. coccoides can positively modulate mucus function in mice through secretion of short-chain fatty acids (SCFAs) and activation of the SCFA receptor Ffar2.

## Results

### Human-derived microbiota prevents mucus defects in mice fed a WSD in a fiber-dependent manner

To investigate whether increased fiber intake in humans could alter the capacity of their gut microbiota to modulate colonic mucus function,

we assessed 67 healthy middle-aged individuals who participated in a recent fiber-intervention trial[32] during which they increased their average daily fiber intake from $22.5 \pm 8.0$ g/day to $36.0 \pm 9.6$ g/day over 12 weeks (Fig. 1A; $p < 0.0001$). The trial was conducted under free-living conditions, and participants increased their fiber intake by consuming foods that were mainly enriched with wheat fiber (insoluble dietary fiber) and oat fiber (soluble fiber)[32]. While no significant difference between baseline daily fiber intake was observed between men and women ($23.59 \pm 9.57$ g/day vs. $21.39 \pm 5.80$ g/day), men increased their fiber consumption on average by 48.6%, while women increased their consumption by 74.6% during the intervention (S1A). Nevertheless, men and women responded equally well during the intervention (S1B). Overall, bacterial 16S rRNA gene sequencing analysis revealed intraindividual alterations in the microbial community when the participants switched from their habitual diet (HD) to the high-fiber diet (HF), but no significant changes were observed when assessing the whole cohort ($p = 0.452$, S1C).

All 67 participants were ranked based on their improvement in waist-to-hip ratio, body mass index and free fat mass as well as cholesterol, triglyceride and glucose levels. From the ten best responders, compositional changes in the fecal bacterial community, based on Bray–Curtis dissimilarity and weighted UniFrac distance metric, were quantified. After identifying those individuals with the strongest changes, five donors (Fig. 1A and S1D, E) were then selected for our proof-of-concept study, which aimed to show that fiber-mediated alterations in human gut bacteria composition can affect mucus function.

Shifts in stool bacteria composition were observed in each of the five selected participants when switching from HD to HF diet (Fig. 1B and S1D). We thus pooled HD and HF stool samples from those individuals to perform fecal microbiota transplantation (FMT, repeated after 5 days) into antibiotic-treated and bacteria-depleted mice, which, in contrast to germ-free mice, do not have an understimulated intestinal immune system[33]. Furthermore, microbiota-transplanted mice received either a control chow diet or a low-fiber WSD to disentangle the interplay between diet, transplanted microbiota and colonic mucus function, which was investigated on viable colonic explants using an ex vivo perfusion chamber (Fig. 1C).

Mucus growth is an important aspect of mucus function, which acts to replenish the mucus layers and to generate a constant luminal-directed flow away from the host epithelium. After receiving the human HD microbiota, the mucus growth rate in the distal colon was significantly reduced in mice fed a WSD compared to mice fed a chow diet ($p_{adj} < 0.001$, Fig. 1D). This indicates that the participants' habitual microbiota is not able to maintain proper colonic mucus function upon WSD feeding. Remarkably, in mice receiving the high-fiber microbiota transplant, no reduction in mucus growth rate was observed ($p_{adj} = 0.93$), despite being fed a WSD. Furthermore, both mouse groups receiving the HF microbiota displayed a higher mucus growth rate than the HD-WSD group ($p_{adj} = 0.001$ and $p_{adj} < 0.001$), suggesting that high-fiber consumption shifted the human microbial community toward a configuration with a higher capacity to maintain mucus growth under unfavorable low-fiber conditions. Accordingly, mouse diet ($p_{adj} = 0.01$), donor microbiota ($p_{adj} < 0.001$) and the interaction of both ($p_{adj} < 0.01$) contributed to the variation in mucus growth rate.

Interestingly, the protective effect of the HF microbiota transplant was specific for mucus function, as mucus thickness (Fig. 1E) and colon length (Fig. 1F) were not affected by the microbiota-diet interaction ($p_{adj} > 0.05$). Moreover, neither expression of the main mucus protein Muc2 nor the intestinal mucosal host defense proteins regenerating islet-derived protein 3-gamma (Reg3γ), deleted in malignant brain tumors 1 (Dmbt1), mouse beta defensin 3 (mBD3), mouse beta defensin 4 (mBD4) and LY6/PLAUR domain containing 8 (Lypd8), displayed a similar microbiota-diet dependency (Fig. 1G). In contrast, the

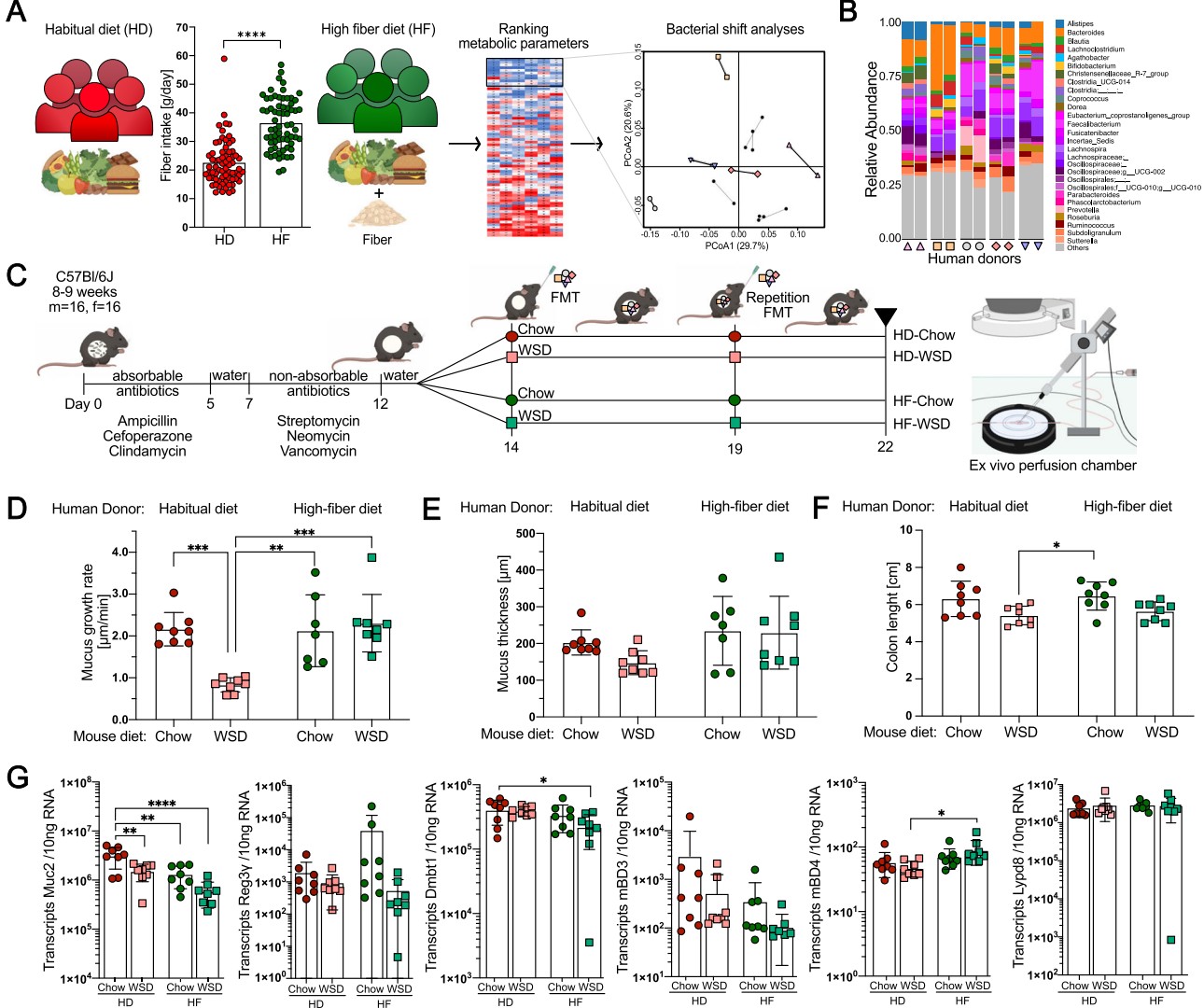

**Fig. 1 | Human-derived microbiota prevent mucus defects in mice fed a Western-style diet (WSD) in a fiber-dependent manner.** HD Habitual diet, HF High-fiber diet, m male, f female. **A** Selection of participants from a previously published intervention study[32], in which participants increased their dietary fiber intake for 12 weeks. Participants ($n = 67$) were ranked based on their improvement in waist-to-hip ratio, body mass index, free fat mass, cholesterol, triglyceride, and glucose levels (heatmap: rows correspond to individual participants, columns correspond to each metabolic parameter). From the top ten responders, shifts in bacterial composition, based on Bray–Curtis and weighted UniFrac distance metric, were calculated, and after combination with the metabolic score, 5 participants within the top 6 responders were selected. Statistical significance was determined by the Wilcoxon matched-pairs signed-rank test. **B** Shift in gut bacteria community structure of the 5 human participants selected as donors for microbiota transplantation before and after HF intervention. **C** Schematic representation of the human-to-mouse FMT experiment using antibiotic-treated mice. Following the first FMT, mice were fed a standard chow diet or a WSD ($n = 8$ mice/group). **D** Mucus growth rate and **E** mucus thickness of the inner colonic mucus layer. **F** Colon length. **G** Absolute quantification of host defense protein/peptide transcripts in distal colon; Statistical significance in D-G was determined by 2-way ANOVA and Tukey's multiple comparison test within microbiota transplant groups. Data in A and D–G are presented as mean ± SD with $p_{(adj)} < 0.05$ (*), $p_{(adj)} < 0.01$ (**), $p_{(adj)} < 0.001$ (***) and $p_{(adj)} < 0.0001$ (****) considered statistically significant. All $P$ values are two-sided. Parts of A and C were created with BioRender.com. Source data are provided as a Source Data file.

expression of Muc2 in fact decreased upon the HF transplant, further indicating that the protective effect is specific for mucus function and that measurement of Muc2 expression is not optimal for full characterization of this post-translational process.

### Relative abundance of *Blautia* correlates with mucus function

Distinct mouse gut microbiota configurations can protect against diet-induced colonic mucus defects[30]. To better understand the influence of human-derived microbiota on mucus function, we analyzed the fecal bacterial composition in the transplanted mice by 16S rRNA gene sequencing. While the baseline bacterial community of the mice was similar for all groups, FMT resulted in a strong shift and replacement of mouse bacterial taxa (Fig. 2A, B). Furthermore, we were able to verify

the presence of similar core bacterial genera in both humans and transplanted mice (S2A, B). Accordingly, despite lower counts, no significant difference in alpha diversity, measured by observed species and Shannon index (S2C), was detected between the human donor samples and the transplanted mice fed a chow diet. In contrast, transplanted mice fed a WSD had significantly lower alpha diversity when compared to the human donor samples, independent of the human donor group. Moreover, mice transplanted with the HD microbiota and fed a WSD had lower observed species counts than mice transplanted with the same microbial community but fed a chow diet. Furthermore, principal coordinate analysis using weighted Uni-Frac and Bray–Curtis dissimilarity matrix (S2D) indicated gut bacterial clustering based on mouse diet, while Bray–Curtis analysis also

identified clustering based on human donor transplant ($p = 0.001$ for HD-Chow vs. HF-Chow and $p = 0.004$ for HD-WSD vs. HF-WSD).

At the phylum level (Fig. 2C), Verrucomicrobiota were more abundant in both WSD groups compared to chow groups ($p_{adj}=0.002$ for HD-WSD vs. HD-Chow, $p_{adj} = 0.014$ for HD-WSD vs. HF-Chow, $p_{adj} = 0.003$ for HF-WSD vs. HD-Chow, and $p_{adj} = 0.019$ for HF-WSD vs. HF-Chow), while Actinobacteriota were significantly reduced in the HD-WSD group when compared to both mouse groups fed a chow diet ($p_{adj}=0.020$ for HD-WSD vs. HD-Chow and $p = 0.008$ for HD-WSD vs. HF-Chow).

At the genus level, the relative abundance of *Bacteroides* (Fig. 2D) was highest in the chow-fed mice transplanted with HF microbiota when compared to any of the other three groups ($p_{adj} < 0.0001$). Furthermore, both WSD-fed mouse groups showed a significant increase in the relative abundance of *Akkermansia* when compared to the HD-Chow group ($p_{adj} = 0.007$ for HD-WSD vs. HD-Chow, $p_{adj} = 0.011$ for HF-WSD vs. HD-Chow). Likewise, *Bilophila* was also increased in both WSD-fed mouse groups ($p_{adj} = 0.007$ for HD-WSD vs. HD-Chow, $p_{adj} < 0.0001$ for HD-WSD vs. HF-Chow, and $p_{adj} = 0.015$ for HF-WSD vs. HF-Chow), while *Enterococcus* was more abundant in the HF-transplanted groups when compared to the HD-transplanted mice, independent of mouse diet ($p_{adj} < 0.0001$). Remarkably, the relative abundance of *Blautia* was reduced in HD-transplanted mice fed a WSD when compared to both chow-fed groups ($p_{adj} = 0.003$ and $p_{adj} = 0.0006$), which was not observed in HF-mice fed a WSD ($p_{adj} = 0.531$ and $p_{adj} = 0.072$), thereby reflecting the observed mucus growth rate pattern (Fig. 1D). Indeed, when correlating the abundances of the 30 most abundant bacterial genera with the measured mucus growth rate, *Blautia* displayed the strongest positive correlation with the mucus growth rate, independent of whether centered log ratio (CLR) transformed abundance ($r_S = 0.4787$; $p = 0.0064$; $p_{(adj)} = 0.0680$) or relative abundance ($r_S = 0.493$; $p = 0.0049$; $p_{(adj)} = 0.0840$) was used (Fig. 2E, S2E and Supplementary Data 2). Moreover, the relative abundance of *Blautia* increased in the stool samples from the human participants after 12 weeks of high-fiber intervention ($p = 0.038$, Fig. 2F), thus raising the hypothesis that *Blautia* may be a diet-dependent commensal that improves colonic mucus function.

## Human-derived high-fiber microbiota ameliorate intestinal infection

While some microbial taxa can be beneficial for mucus function, distinct members of the gut bacteria, including WSD-increased *Akkermansia* (Fig. 2D), can switch from fiber degradation toward host mucin consumption through the production of carbohydrate-active enzymes (CAZy)[34] under low-fiber conditions[21]. When testing the microbial CAZy activity of cecal content from the transplanted mice against different fiber- or mucus-derived glycan substrates, we detected that the combination of chow diet and HF transplant led to increased activity of β-glucosidase (HD-Chow: $p_{(adj)} = 0.0079$, HD-WSD: $p_{(adj)} < 0.0001$, HF-WSD: $p_{(adj)} < 0.0001$) and β-xylosidase (HD-Chow: $p_{(adj)} = 0.023$, HD-WSD: $p_{(adj)} = 0.0006$, HF-WSD: $p_{(adj)} = 0.0026$), which are known to primarily act on dietary fiber (Fig. 3A). Moreover, our observation that both the human microbiota and mouse diet affected the activity of β-glucosidase (donor microbiota: $p = 0.0121$, mouse diet: $p < 0.0001$, interaction: $p = 0.0306$) and β-xylosidase (donor microbiota: $p = 0.0153$, mouse diet: $p = 0.0007$, interaction: $p = 0.0889$) indicates that increased fiber consumption by the human donors leads to higher fiber-degradation capacity by their microbiota in the presence of dietary fiber. This interpretation was further supported by dietary fiber-targeting α-galactosidase activity, which was highest in the HF-transplanted group fed a chow diet (donor microbiota: $p = 0.0255$, mouse diet: $p = 0.0625$, interaction: $p = 0.6441$).

In contrast to the enzymes that primarily target dietary glycans, the activity of sialidase, which predominantly targets terminal sialic acid residues in mucin glycan structures, increased upon WSD feeding in both transplant groups (HD: $p_{(adj)} = 0.0012$, HF $p_{(adj)} = 0.0167$). Sialidase activity was mainly driven by mouse diet but not human donor microbiota (donor microbiota: $p = 0.495$, mouse diet: $p < 0.0001$, interaction: $p = 0.4752$). α-fucosidase activity, which targets terminal fucose residues in mucus glycans, was not affected by diet or microbial transplants, whereas the HF transplant prevented the increase in mucus-active β-N-acetyl-glucosaminidase activity that was observed in HD-transplanted mice upon WSD feeding ($p_{(adj)} = 0.0039$) (Fig. 3A). Taken together, in addition to improving the colonic mucus growth rate, increased fiber consumption by the human donors results in their microbiota favoring fiber over mucin glycan consumption while maintaining high sialidase activity during WSD feeding (Fig. 3B).

Increased microbial sialidase activity during WSD feeding will release terminal sialic acid from mucus, and it was recently shown that the intestinal mouse pathogen *Citrobacter rodentium* can sense and metabolize sialic acid, thereby directing it toward the intestinal mucosa to promote colonization and infection[35]. Moreover, microbial mucus degradation under a low-fiber diet has been shown to promote lethal colitis by *C. rodentium*[21]. We therefore investigated whether the HF microbiota-mediated improvement in mucus growth rate could ameliorate intestinal infection by *C. rodentium* under WSD feeding when compared to HD microbiota (Fig. 3C). Bacteria-depleted mice were orally infected with *C. rodentium* 8 days after the first human microbiota transplant, which was the timepoint at which we previously observed differences in the mucus growth rate (Fig. 1D). Mice were then examined 1–7 days post infection while the infection was known to be still active to focus on early *C. rodentium* colonization rather than host-mediated pathogen clearance. Indeed, we observed that mice transplanted with HF microbiota displayed lower initial colonization and continued reduced levels of *C. rodentium* colony-forming units (CFUs) when compared to the HD-transplanted mice (Fig. 3D). Similarly, HF-WSD mice showed no reduction in body weight (Fig. 3E) and a higher number of mucus-filled goblet cells per crypt in the HF-transplanted group ($p = 0.046$; Fig. 3F). Taken together, these findings suggest that fiber-dependent and microbiota-mediated improvement in mucus growth can limit intestinal infection, which is consistent with previous findings showing that the presence of an intact mucus layer is crucial in limiting *C. rodentium* infection[10]. Seven days post infection, CFUs of *C. rodentium* were still lower in the cecum of HF-transplanted mice ($p = 0.008$; Fig. 3G), while the expression of Muc2, Reg3γ, Dmbt1, mBD3, mBD4 and Lypd8 was not altered between HD- and HF-transplanted mice infected with *C. rodentium* (S3A). Histology of the distal colon revealed no major difference in mucosal integrity (S3B) or crypt length (S3C) and colon length was similar between the two groups (S3C), suggesting that the lower *C. rodentium* counts are not caused by inflammation but could be due to mucus function.

Analysis of the colonic bacterial composition in the transplanted and infected mice again confirmed successful transfer of the predominant bacterial genera from humans to mice (S3D, E), which resulted in distinct clustering of the baseline and post-FMT samples ($p = 0.001$; Fig. 3H) and a replacement of the inherent mouse bacteria (Fig. 3I). After FMT and *C. rodentium* infection, the bacterial community remained stable (S3F) and distinct between the HD and HF transplant groups at least until day 5 post infection, which was 13 days after the first FMT (Fig. 3J and S3G).

Motivated by the earlier association between the relative abundance of *Blautia* and mucus function (Fig. 2E), we analyzed the abundance of this taxon in the stool of the infected mice on day 1 after infection (day 8 after the initial transplant, at which we previously detected the difference in mucus growth rate between the WSD-fed mice (Fig. 1D)). We observed a higher abundance of *Blautia* in the HF-transplanted mice than in the HD-transplanted group ($p = 0.026$, Fig. 3K), and interestingly, the abundance of *Blautia* correlated

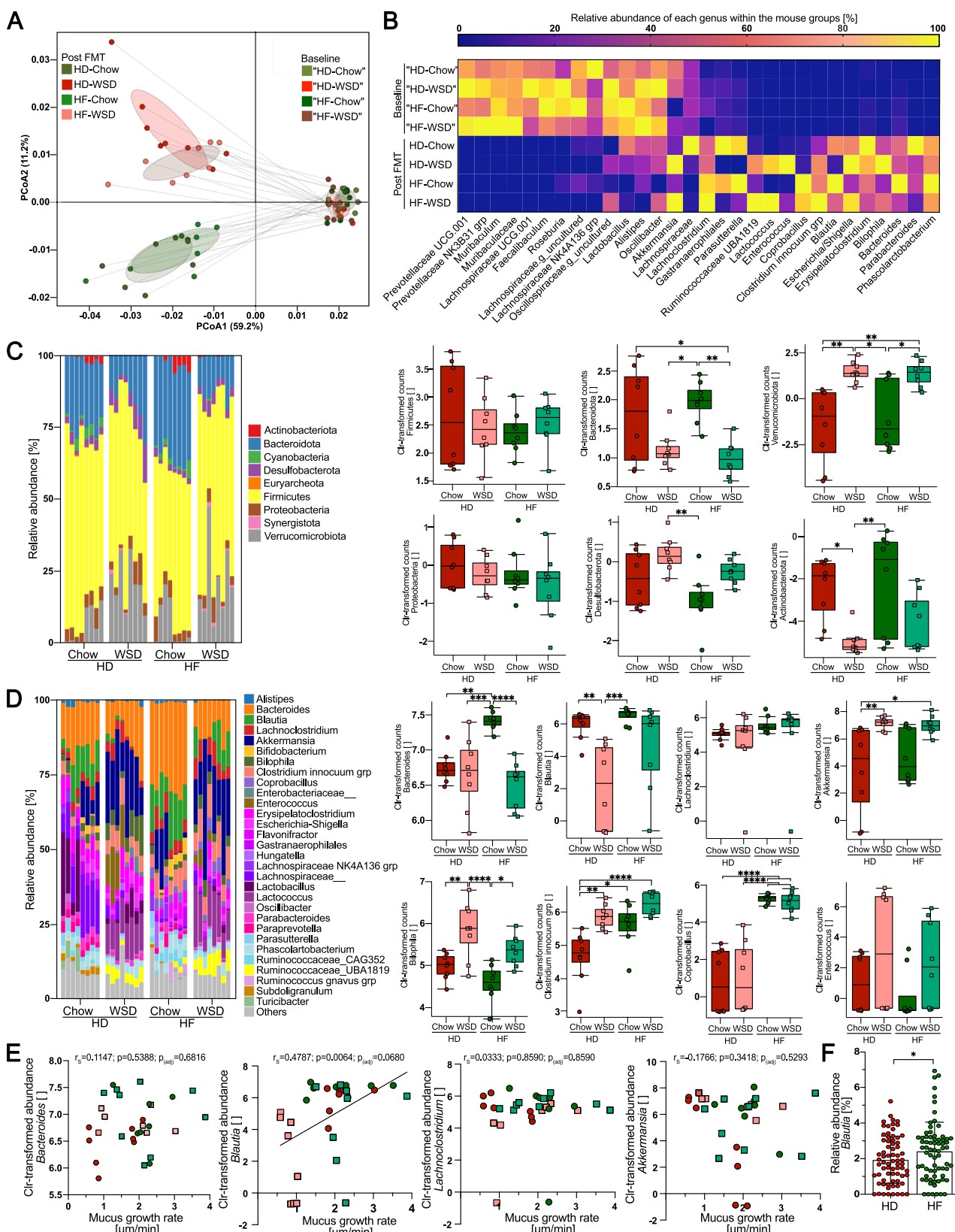

negatively with the fecal *C. rodentium* counts at this early timepoint after infection (Fig. 3L), which corroborates the possibility that *Blautia* could improve mucosal barrier function. However, since the baseline bacterial community before antibiotic treatment differed between the two transplantation groups (Fig. 3H), we cannot fully exclude that early-life colonization of the mice contributed to the differential

response to *C. rodentium* infection, despite the antibiotic treatment before FMT.

***Blautia coccoides* improves mucus function in WSD-fed mice**
*Blautia* has been identified as a mucus-associated bacterium in human and mouse colon[36,37], and recent findings[38] indicate that most

**Fig. 2 | Relative abundance of the gut commensal *Blautia* correlates with mucus function. A** Weighted UniFrac PCoA and (**B**) relative abundance of bacterial genera before (baseline) and after (termination) human-to-mouse FMT. **C** Abundance of bacterial taxa in fresh stool samples from mice transplanted with habitual diet (HD)- or high-fiber diet (HF)-derived human microbiota and fed a chow or Western-style diet (WSD) at phylum and (**D**) genus level (*n* = 8 mice/group). Relative abundance taxa plots (left) display the top 30 bacterial taxa. Boxplots (right) display Centered log-ratio (CLR) transformed abundance counts of selected taxa, with data presented as median with upper and lower quartiles. Statistical significance was determined by 2-way ANOVA and Tukey's multiple comparison test. **E** Spearman correlation analysis between mucus growth rate in the distal colon and centered log ratio (Clr)-transformed relative abundance of selected genera. Data was tested for normal distribution using the D'Agostino & Pearson test and *P* values were adjusted for multiple comparisons using the Benjamini–Hochberg procedure. For adjusted *P* values in (**D**) and (**E**), $p_{(adj)} < 0.05$ (*), $p_{(adj)} < 0.01$ (**), $p_{(adj)} < 0.001$ (***) and $p_{(adj)} < 0.0001$ (****) were considered statistically significant. **F** Relative abundance of *Blautia* in human participants (*n* = 67) before and after 12 weeks of high-fiber intervention. Data is presented as mean ± SD and statistical significance was determined by the Wilcoxon matched-pairs signed-rank test, with $p < 0.05$ (*) considered statistically significant. All *P* values are two-sided. Source data are provided as a Source Data file.

identified *Blautia* species display lower prevalence in industrialized populations when compared to the traditionally living Hadza hunter-gatherers of Tanzania (S4A), who consume high amounts of dietary fiber[39]. While our 16S rDNA sequencing approach did not allow resolution of the *Blautia* genus at the species level, blast searches of the *Blautia* amplicon sequence variants (ASVs) in the transplanted mice suggested several potential species (Supplementary Data 1), including *Blautia coccoides*, the type species of this genus[40,41] that has previously been detected in the human gut[42]. To test our hypothesis that *Blautia* is able to stimulate mucus growth, we switched conventionally raised mice to a WSD and supplemented one group with *B. coccoides* for 5 weeks through drinking water, while the remaining mice received bacterial growth media control in the drinking water (Fig. 4A and S4B). Supplementation with *B. coccoides* caused a significant increase in the mucus growth rate (*p* = 0.038) (Fig. 4B), but not mucus thickness (*p* > 0.999) (S4C), in the distal colon when compared to the control group. Moreover, by using confocal microscopy on viable distal colon explants (Fig. 4C), we further observed that the mucus of the *B. coccoides*-treated mice was less penetrable to bacteria-sized beads when compared to the control treatment, thereby confirming the detrimental effect of a low-fiber diet on mucus integrity[21–23,30]. Accordingly, quantification and localization of penetrating beads, determined by measuring distances of individual beads to the colonic epithelium, identified that *B. coccoides* treatment resulted in a significant increase in bead distance from the intestinal epithelium (Fig. 4D, *p* < 0.0001). Moreover, within the mucus of individual mice, *B. coccoides* supplementation also reduced the fraction of beads penetrating the area within 10 μm (*p* = 0.016) from the colonic epithelium (Fig. 4E), indicating that *B. coccoides* could reduce bacterial penetration into the area with the highest risk for epithelial contact.

*B. coccoides* is an anaerobic bacterium that subsequently may not survive for an extended period in autoclaved drinking water, and consequently, our more natural supplementation approach does not allow conclusions of whether the mucus growth-inducing effect needed viable bacteria. Hence, to better control viability and the number of supplemented bacteria, we repeated *B. coccoides* application through oral gavage (Fig. 4F). Supplementation with viable *B. coccoides* caused a significantly higher mucus growth rate (*p* = 0.0022) than heat-killed *B. coccoides* (Fig. 4G), thereby confirming that colonic mucus growth is induced by viable bacteria, either directly or through modulation of the colonic microbial community (S4D), again without affecting mucus thickness (S4E). Supplementation with viable *B. coccoides* also increased the distance of bacteria-sized beads from the intestinal epithelium compared to heat-killed bacteria (*p* < 0.0001, Fig. 4H, I), albeit without significantly affecting the fraction of beads penetrating the mucosa-adherent layer (Fig. 4J). Interestingly, the mucus-modulating capacity seemed species-specific, as the mucus growth rate did not significantly increase (*p* = 0.3095) after oral gavage with a different *Blautia* species, *Blautia wexlerae*, when compared to the respective heat-killed control group (S4F, S4G).

## *Blautia coccoides* partly ameliorates *C. rodentium* infection

To investigate whether *B. coccoides*-induced mucus growth also recapitulates the amelioration of *C. rodentium* infection, we fed conventionally raised mice a WSD, with or without *B. coccoides* supplementation through drinking water, and infected both groups with *C. rodentium* (Fig. 5A). We performed the infection 8 days after starting *B. coccoides* supplementation, as we previously observed microbiota-induced mucus growth 8 days after human microbiota transplantation (Fig. 1), and we observed that human microbiota reduced *C. rodentium* CFUs at this timepoint after infection (Fig. 3D). CFUs of *C. rodentium* in the stool of the infected mice declined faster in the *B. coccoides*-treated mice than in the control group (Fig. 5B), and as expected, the *B. coccoides* supplement resulted in a strongly increased colonic mucus growth rate (*p* = 0.007), again without affecting mucus thickness (Fig. 5C). Seven days post infection, however, only a tendency for lower *C. rodentium* CFUs was observed in the colon, cecum, spleen and liver (S5A) of the *B. coccoides*-treated mice, suggesting that *B. coccoides*-induced mucus growth reduced the *C. rodentium* load early during infection but did not lead to faster clearance of the pathogen overall. Mucosal integrity (S5B), number of mucus-filled goblet cells per crypt and colon length (S5C) were similar between the two groups, as was the expression of colonic host defense molecules, including Muc2 (Fig. 5D) and the change in body weight (S5D). Moreover, the baseline bacterial composition before *B. coccoides* supplementation and *C. rodentium* infection was also similar between the two mouse groups (Fig. 5E), ruling out any difference in *C. rodentium* susceptibility based on earlier bacterial colonization. Thus, *B. coccoides* supplementation led to a reduction in *C. rodentium* load early during the infection, but to a weaker extent than was observed in the HF-transplanted mice (Fig. 3). It is therefore likely that the contribution of other microbial members is needed for more robust protection against *C. rodentium*.

## *Blautia coccoides* stimulates mucus growth through the production of SCFAs

To gain a better understanding of how *B. coccoides* stimulates mucus function, we performed high-throughput global targeted metabolomics profiling of the colonic mucus from WSD-fed mice supplemented with *B. coccoides* through drinking water. While partial least squares-discriminant analysis (PLS-DA) and unsupervised hierarchical cluster analysis of the 147 reliably detected metabolites only identified a moderate overall metabolome separation between the mucus of the *B. coccoides*-supplemented mice and the control group (Fig. 6A), several differentially abundant metabolites were identified (Fig. 6B). Among them, the SCFAs acetate (*p* = 0.0095) and pentanoate (*p* = 0.0381) as well as the medium-chain fatty acids hexanoate (*p* = 0.0190) and heptanoate (*p* = 0.0381) were increased in the mucus of *B. coccoides*-supplemented mice, while the differences in propionate (*p* = 0.0667) and butyrate (*p* = 0.1143) levels did not reach statistical significance (Fig. 6C). SCFAs have been attributed to improved mucosal barrier function[43,44,] and correlation analysis of these acids revealed that the signal for propionate correlated significantly with the colonic mucus growth rate (Fig. 6D).

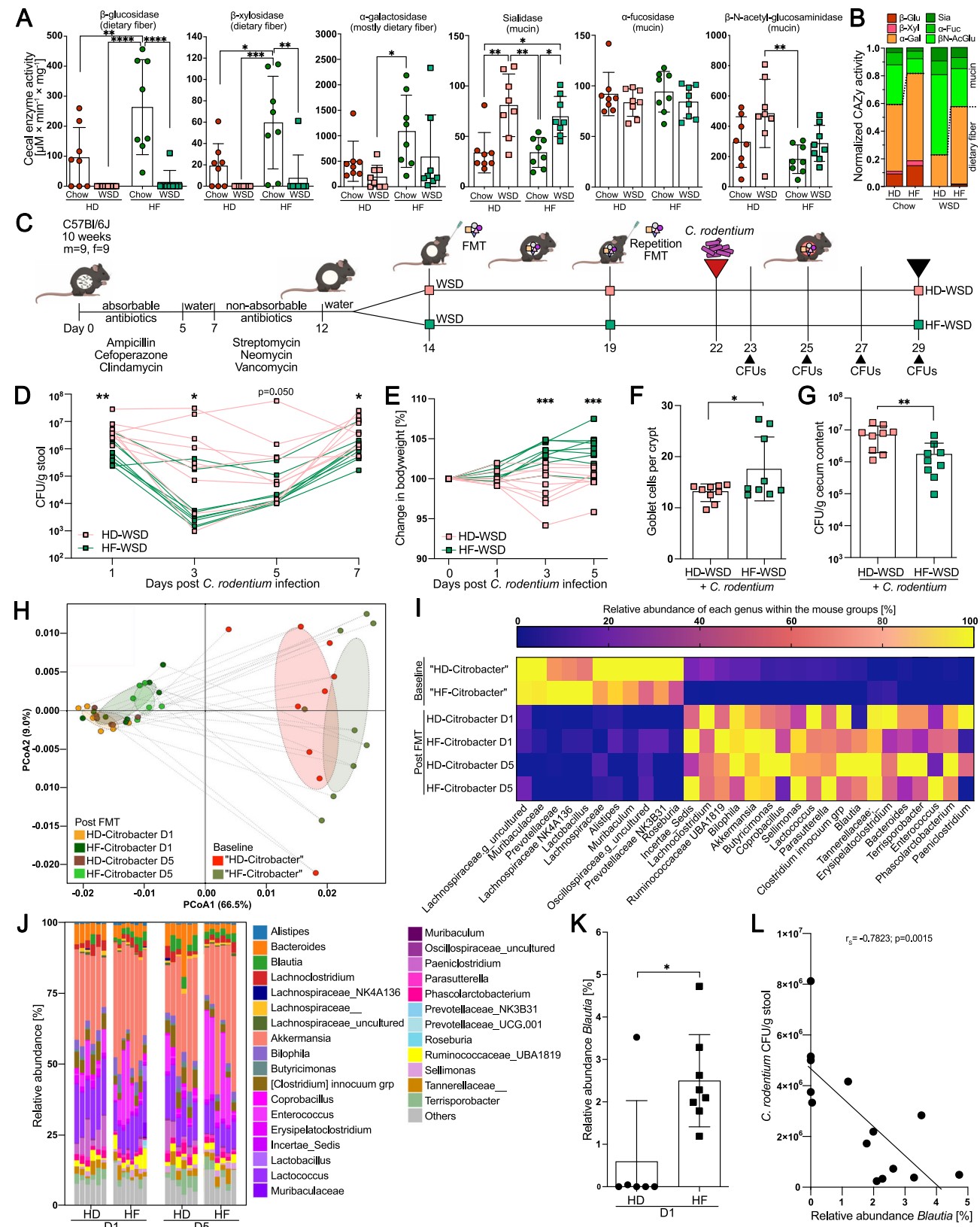

Since we previously observed a correlation between *Blautia* abundance and mucus growth in the mice transplanted with the human microbiota (Fig. 2E), we consequently aimed to test whether SCFAs could have contributed to mucus growth in these mice as well. We therefore performed targeted metabolomics profiling on the cecal content of the transplanted mice and compared the metabolite profile

of the mouse groups with a normal average mucus growth rate (HD-Chow, HF-Chow, and HF-WSD) to the group with a reduced average mucus growth rate (HD-WSD). Despite PLS-DA identifying significant clustering based on mucus function (S6A), hierarchical cluster analysis by using Euclidian distance measurement identified mouse diet as the dominating factor, which could have masked any potential metabolite

**Fig. 3 | Human-derived high-fiber microbiota ameliorate intestinal infection.**
**A** Enzymatic activity of carbohydrate-active enzymes (CAZy) in the cecal content of human microbiota-transplanted mice ($n = 8$ mice/group) against various carbohydrate structures. Statistical significance was determined with 2-way ANOVA and Tukey's multiple comparison test. **B** Normalized CAZy activity of enzymes primarily targeting dietary fiber (red/orange) and mucus glycans (green). **C** Schematic representation of the human FMT and *Citrobacter rodentium* infection experiment ($n = 9$ mice/group). **D** Change in CFUs of *C. rodentium* in stool samples and (**E**) change in body weight after *C. rodentium* infection. **F** Number of goblet cells per crypt in the distal colon and (**G**) CFUs of *C. rodentium* in the cecum 7 days post infection ($n = 9$ mice/group). (**H**) Weighted UniFrac PCoA and (**I**) relative abundance of bacterial genera before (baseline) ($n = 9$ mice/group) and after human-to-mouse FMT (HD: $n = 6$ mice; HF: $n = 8$ mice). **J** Relative genus abundance in cecal content of the transplanted and infected mice on day 1 (D1) and day 5 (D5) post infection; **K** Relative abundance of *Blautia* at D1 post infection; Statistical significance was determined with the Mann–Whitney $U$ test. **L** Spearman correlation between the relative abundance of *Blautia* and *C. rodentium* CFUs in stool at D1 post infection. Statistical significance was determined using the Mann–Whitney $U$ test for (**D**)–(**G**). For (**D**–**H**) and (**K**), data are presented as mean ± SD, and normal distribution was tested with the D'Agostino & Pearson test. $p < 0.05$ (*), $p < 0.01$ (**), $p < 0.001$ (***) and $p < 0.0001$ (****) were considered statistically significant. All $P$ values are two-sided. HD habitual diet, HF high-fiber diet. Parts of (**C**) were created with BioRender.com. Source data are provided as a Source Data file.

profile that depended on mucus function (S6B). Accordingly, correlation of the colonic mucus growth rate from the transplanted mice with their respective cecal levels of SCFAs did not identify any significant correlations, while the medium-chain fatty acids hexanoate and heptanoate correlated significantly with mucus growth (S6C). To thus prevent the strong effect of diet, we compared only the mouse groups fed a WSD and again observed a significant correlation ($p = 0.0231$) between propionate levels and mucus growth rate (Fig. 6E), while the correlation between hexanoate and mucus growth rate remained significant as well ($p = 0.0223$). In contrast, no significant correlations were observed for acetate, butyrate, pentanoate, or heptanoate, thereby unveiling propionate as the most promising microbial metabolite that may be involved in mucus growth induction.

To experimentally validate the positive associations between propionate and mucus growth, we divided viable mouse distal colon tissues from conventionally raised mice fed a WSD into two pieces and stimulated the first piece ex vivo with a buffer control, while the second piece from the same mouse was stimulated with propionate. Confirming the previously identified association between propionate and the mucus growth rate, we found that propionate stimulated mucus growth in a concentration-dependent manner (Fig. 6F).

To next establish whether mucus-stimulating metabolites can be produced by *B. coccoides*, we continued to test whether *B. coccoides* supernatant can induce mucus growth. Therefore, mouse distal colon tissue was divided into two pieces as described above, and one piece was stimulated with brain heart infusion-supplemented (BHI-S) media, while the second piece was stimulated with the supernatant of a *B. coccoides* culture grown in BHI-S. Colonic tissue stimulated with the *B. coccoides*-derived supernatant displayed a significantly increased mucus growth rate ($p = 0.0022$) when compared to the media control (Fig. 6G). Moreover, a stimulatory effect was also observed in the presence of a complex microbial community, where in vitro supplementation of *B. coccoides* to the cecal content of mice fed a WSD led to significantly increased mucus growth after incubation ($p = 0.026$) when compared to the supernatant of the cecal community alone (Fig. 5H). Consequently, these results confirm that *B. coccoides* can induce mucus growth through the production of metabolites, even in the presence of commensal microbiota.

Surprisingly, relative quantification of SCFAs in the mucus-stimulating supernatant of *B. coccoides* revealed that the bacterium can produce significant amounts of acetate but neither propionate nor butyrate (Fig. 6I, S6D) under the conditions used. Similarly, the cecal community supplemented with *B. coccoides* contained higher amounts of acetate than the non-supplemented community (Fig. 6J, S6E), suggesting that *B. coccoides* may induce mucus growth through the production of acetate. Indeed, when stimulating the colonic tissue of WSD-fed mice ex vivo, we observed that acetate, which was strongly increased in the mucus of the *B. coccoides*-supplemented mice, stimulated mucus growth at higher concentrations (Fig. 6K).

Remarkably, however, *Blautia* spp. encode genes of the propanediol pathway for propionate formation from deoxy sugars, such as fucose and rhamnose[45]. When consequently changing the in vitro culturing conditions to Gifu Anaerobic Broth (GAM) media supplemented with fucose, *B. coccoides* produced substantial amounts of acetate and propionate (Fig. 6L) but no butyrate (S6F). Additionally, the supplementation of rhamnose to GAM media led to *B. coccoides*-mediated propionate production (S6G), although at lower amounts than in the fucose supplemented media.

Lastly, SCFAs are mainly sensed by the host through Free fatty acid receptors 2 and 3 (Ffar2/3 Ffar2 = GPR43; Ffar3 = GPR41); also called G-protein coupled receptors 43 and 41 (GPR43/41)), which are expressed by colonic epithelial cells[43]. Since acetate is the most selective SCFA for Ffar2 and propionate is its most potent activator[46], we hypothesized based on the mucus-stimulation results that Ffar2 may be the receptor that senses SCFAs, thereby inducing mucus growth. Ex vivo stimulation of colonic tissues with acetate or propionate in the presence of a commercial Ffar2 inhibitor abolished the stimulatory effect of both SCFAs ($p = 0.0159$ and $p = 0.0079$; Fig. 6M). In contrast, a commercial Ffar2 agonist led to a significant increase in mucus growth when compared to the vehicle control ($p = 0.0079$). Finally, stimulating colonic tissue ex vivo with the supernatant of *B. coccoides* in the presence of the Ffar2 inhibitor significantly reduced the mucus growth rate ($p = 0.026$, Fig. 6N)), thereby mechanistically linking *B. coccoides*-produced SCFAs to Ffar2-mediated mucus growth.

In conclusion, our findings demonstrate that enrichment of the habitual diet with dietary fiber in humans has the potential to alter the gut bacteria configuration, thereby improving microbial interaction with intestinal mucus. Furthermore, we identified *B. coccoides* and the microbial metabolites propionate and acetate as modulators that induce colonic mucus growth by stimulating Ffar2, thereby identifying, to our knowledge, the first mechanistic bacterium-metabolite-mucus functional link in mice.

## Discussion

Intestinal mucus is at the interface between the host and microbiota and thus has a crucial role in protecting the intestinal epithelium against infection, inflammation, and even against toxicity of the transition metal copper[47]. To fulfill this role and to replenish mucus degraded by the gut bacteria from the luminal side, mucus is continuously secreted from goblet cells under homeostatic conditions, resulting in a luminal-directed flow that actively flushes microbes away from the epithelium. Mucus growth is therefore a crucial aspect of intact mucus function that can be used as an indicator of gut health status.

We previously identified a probiotic *Bifidobacterium longum* strain as an inducer of mucus growth in mice[30], but the microbial metabolite(s) mediating this effect remained unknown. Here, we not only identify the gut commensal *Blautia coccoides* as a mucus growth-inducing bacterium but also provide mechanistic insight by demonstrating that the *B. coccoides*-derived metabolites propionate and acetate stimulate mucus secretion through the SCFA receptor Ffar2. As Bifidobacteria are known acetate producers, it is possible that

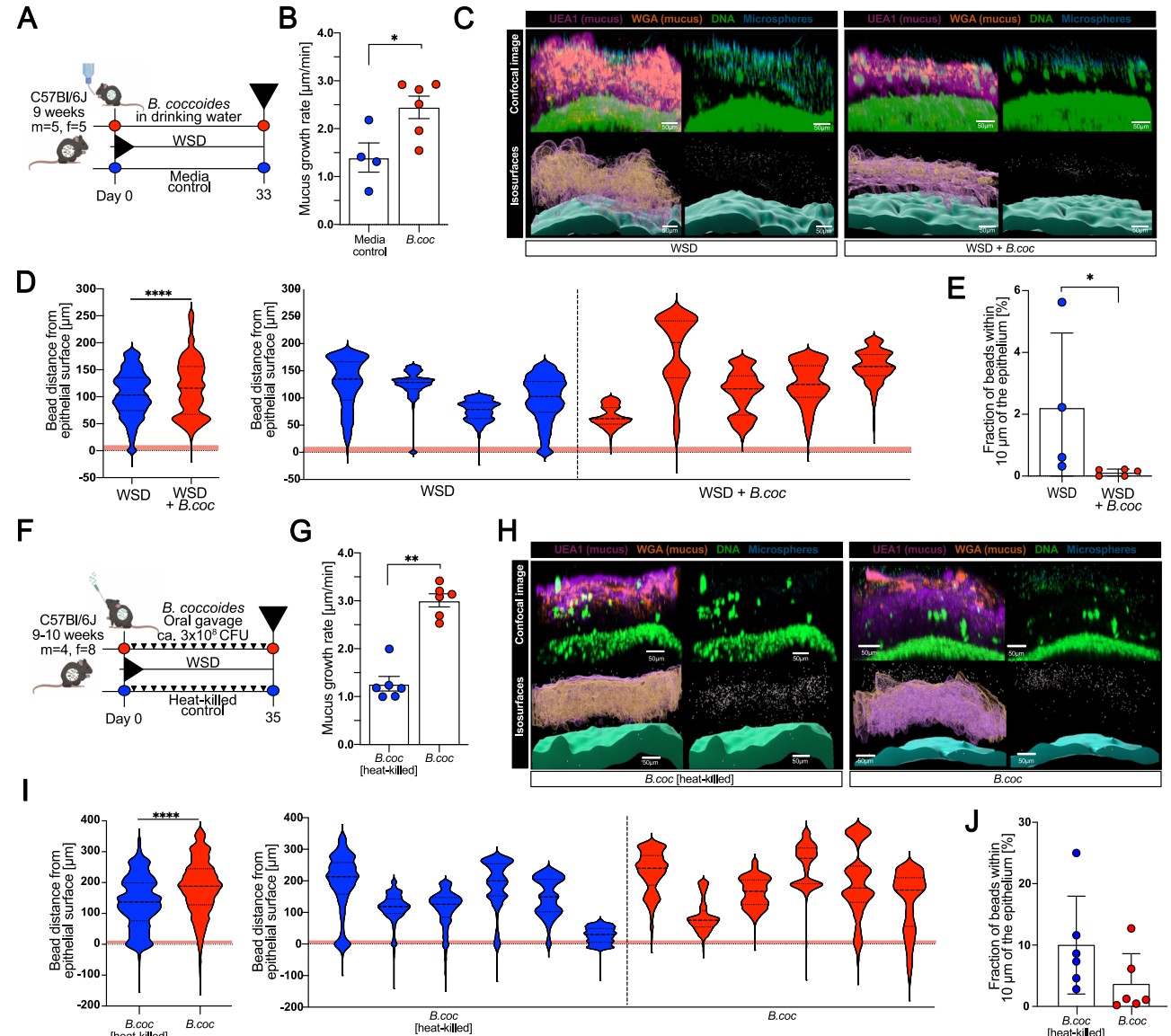

**Fig. 4 | *Blautia coccoides* improves mucus function in WSD-fed mice.**
**A** Schematic representation of *B. coccoides* supplementation through drinking water to WSD-fed mice. Mice were supplemented with media control (*n* = 4 mice) or *B. coccoides* (*n* = 6 mice) for a period of 33 days, whereupon mucus function was investigated. **B** Mucus growth rate of inner colonic mucus layer. **C** Representative confocal images (top) and processed iso-surface images (bottom) from distal colon explants stained for DNA (green), fucosylated mucin glycans (UEA1; purple), sialylated mucin glycans (WGA; orange) overlaid with 1 μm fluorescent microspheres (blue). **D** Position-based microsphere (bead) distribution within mucus layer per group (left) and individual mouse (right), obtained from processed images (2–3 images/mouse). Red shading indicates zone within 10 μm from colonic epithelium. **E** Fraction of microspheres (beads) penetrating red zone (within 10 μm from colonic epithelium) of the mucus layer as indicated in **D**. Values are an average of 2–3 analyzed images/mouse. **F** Schematic representation of *B. coccoides* supplementation through oral gavage of WSD-fed mice. Mice were supplemented with viable or heat-killed *B. coccoides* (*n* = 6 mice/group) through repeated oral gavage over a period of 35 days, whereupon mucus function was investigated. **G** Mucus growth rate of inner colonic mucus. **H** Representative confocal images (top) and processed iso-surface images (bottom) from distal colon explants stained as described for (**C**). **I** Position-based microsphere (bead) distribution within mucus layer per group (left) and individual mouse (right), obtained from processed images (2–4 images/mouse). Red shading indicates zone within 10 μm from colonic epithelium. **J** Fraction of microspheres (beads) that penetrate the red zone (within 10 μm from colonic epithelium) of the mucus layer as indicated in **I**. Values are the average of 2–4 analyzed images per mouse. Data in (**B**), (**G**), (**E**), and (**J**) are presented as mean ± SD. Data in (**D**) and (**I**) are presented as median and quartiles. Normal distribution was tested with the D'Agostino & Pearson test. Statistical significance was determined using the Mann–Whitney *U* test, with *p* < 0.05 (*), *p* < 0.01 (**) and *p* < 0.001 (***) considered statistically significant. All *P* values are two-sided. Parts of (**A**) and (**F**) were created with BioRender.com. Source data are provided as a Source Data file.

*Bifidobacterium longum* also induces mucus growth through acetate production.

Several *Blautia* species have recently been identified to have beneficial effects on host function, including the suppression of *Vibrio cholera* colonization[48], improving inflammatory responses in a DSS mouse model[49], ameliorating obesity and type-2 diabetes[50] and protecting against enteric viral infection[51], despite being a rather low-abundance gut bacterium. The molecular mechanism has not been identified in all cases, but SCFA production was shown to be an important factor for some of these observations.

Metabolite concentrations vary throughout the intestinal tract and differ between the lumen and mucus[37]. Propionate and acetate are common SCFAs that are produced by many different gut bacteria[43], and colonic luminal SCFA concentrations typically occur in the range

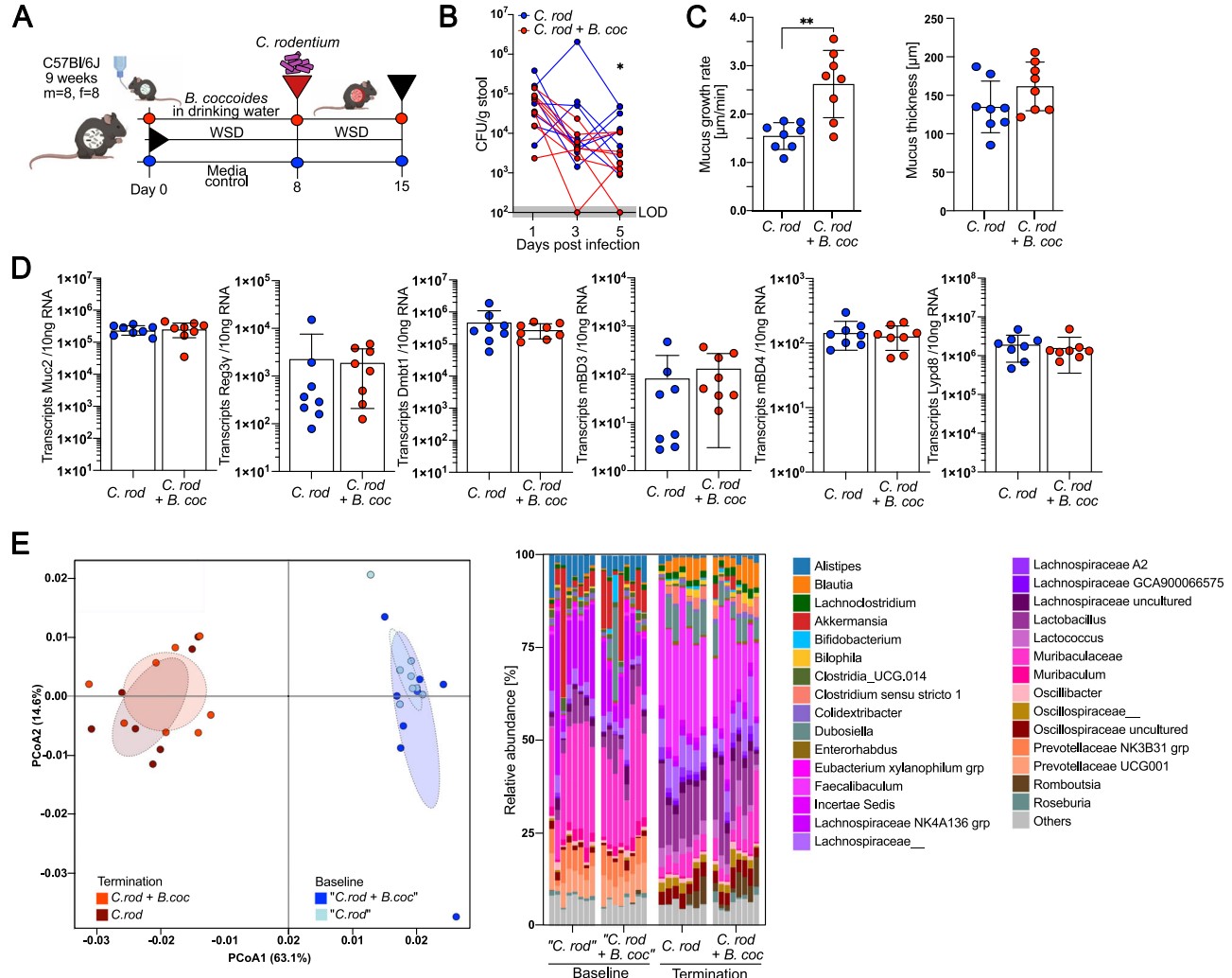

**Fig. 5 | *Blautia coccoides* partly ameliorates *Citrobacter rodentium* infection.**
**A** Schematic representation of the *B. coccoides* supplementation experiment in *C. rodentium*-infected mice. Mice fed a WSD and supplemented with *B. coccoides* or media control through drinking water were infected with *C. rodentium* 8 days after the start of treatment ($n = 8$ mice/group). **B** Monitoring of *C. rodentium* CFUs from stool, LOD = limit of detection. **C** Mucus growth rate and thickness measured 7 days post infection. **D** Absolute quantification of host defense protein/peptide transcripts in the distal colon. **E** Weighted UniFrac PCoA and relative abundance of bacterial genera before (baseline) and after switching to WSD feeding, infecting with *C. rodentium* and supplementing with *B. coccoides* or media control (termination); Data in (**B**–**D**) are presented as mean ± SD and normal distribution was tested with the D'Agostino & Pearson test. Statistical significance was determined by the Mann–Whitney *U* test with $p < 0.05$ (*), $p < 0.01$ (**), and $p < 0.001$ (***) considered statistically significant. All *P* values are two-sided. Parts of (**A**) were created with BioRender.com. Source data are provided as a Source Data file.

between 10 and 100 mM in relative molar ratios of approximately 60-20-20 for acetate-propionate-butyrate, respectively[52]. Since propionate is a more potent activator of Ffar2 than acetate[46] and propionate was more potent in stimulating the mucus growth rate (Fig. 6F, G), the higher concentrations of acetate available in the gut may still be a weaker inducer of mucus growth than propionate, despite its lower concentration.

While we show here that microbial propionate and acetate production can induce mucus growth, the configuration of the gut microbial community at the mucosa, as well as the available substrates, will determine whether the levels of these microbial metabolites will reach sufficient concentrations to affect mucus function. Thus, although we could here identify a specific bacterium and two metabolites that induce colonic mucus growth, the addition of a single bacterium to an established community may not always be sufficient to stimulate mucus growth under all conditions. Correspondingly, we observed that *B. coccoides* was able to produce propionate in vitro in GAM media supplemented with fucose or rhamnose but not in GAM or BHI media alone. Interestingly, fucose is a terminal glycan residue of

colonic mucin glycans[53], and since *Blautia* has been detected in human and mouse mucus[36,37], this allows the possibility that *B. coccoides* consumes mucin-derived fucose that has been liberated either through enzymatic activity of *B. coccoides* or other microbes. Subsequent production of propionate and acetate could then replenish the mucus layer in the colon, thereby providing more growth substrate and at the same time maintaining host barrier function. Such a hypothesis would be in line with previous studies that found increased numbers of mucin-producing goblet cells after supplementing mucin-degrading and SCFA-producing *Akkermansia muciniphila* to high-fat diet-fed mice[54] or with the fucosylation-inducing effect of fucose-degrading *Bacteroides thetaiotaomicron*[55]. Likewise, increased abundance of *Blautia* and increased production of propionate were detected when human fecal microbiota was incubated in vitro in the presence of mucin[56]. Additionally, a recent study proposed that fucose and rhamnose could be liberated from dietary fibers through secreted microbial glycoside hydrolases[37], and we could indeed measure gut microbial fucosidase activity in the mice transplanted with the human-derived microbiota in our study (Fig. 3A).

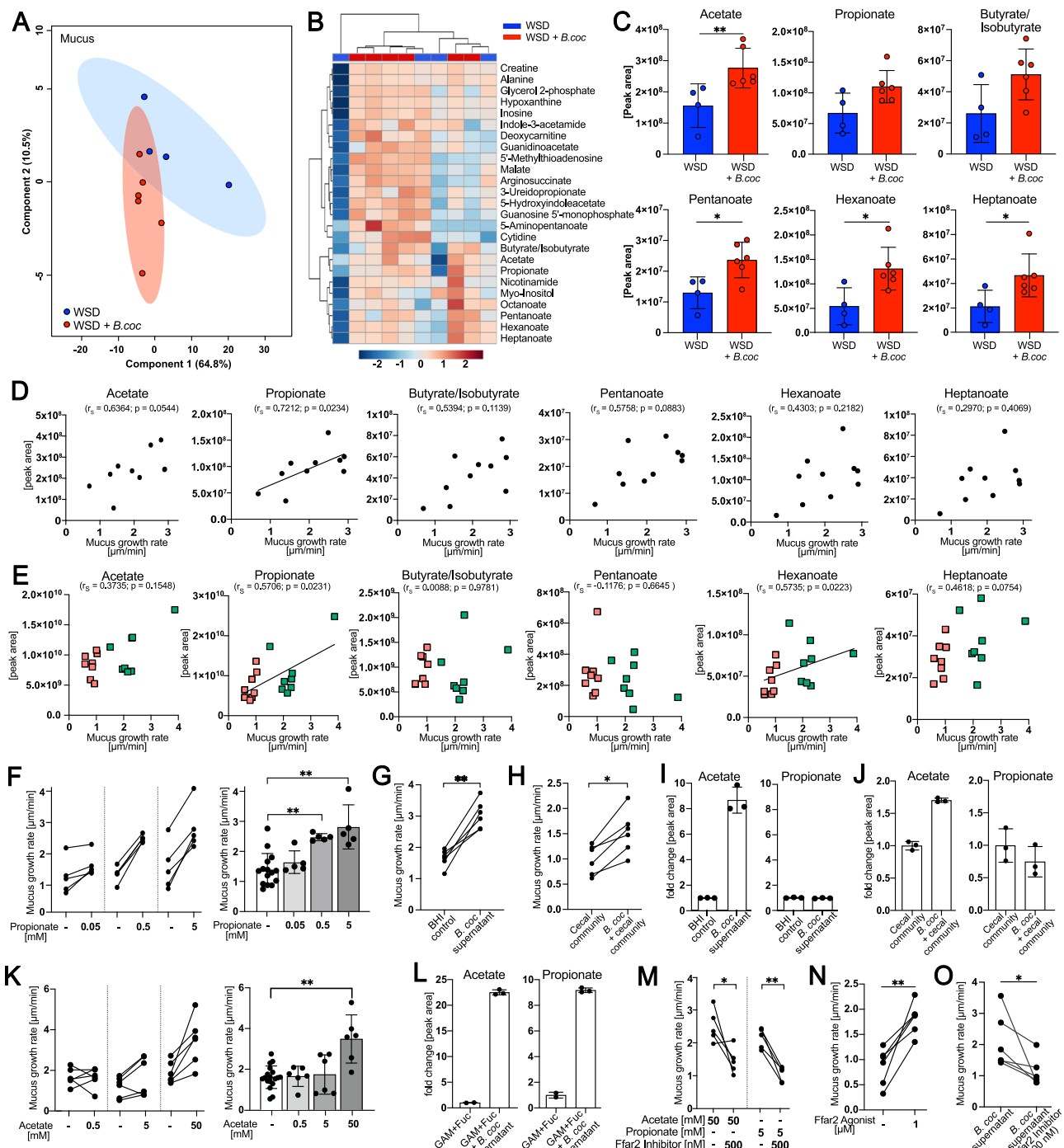

**Fig. 6 | *Blautia coccoides* stimulates mucus growth through short-chain fatty acid production. A** PLS-DA of mucus metabolites. WSD-fed mice (*n* = 10) were supplemented with *B. coccoides* (red, *n* = 6) or control media (blue, *n* = 4) through drinking water. **B** Unsupervised hierarchical cluster analysis using Euclidian distance measurements of the 25 most altered metabolites. Color scale indicates fold change in normalized peak intensity. **C** Peak area of altered SCFAs, hexanoate and heptanoate. **D** Spearman correlation between colonic mucus growth rate and SCFAs, hexanoate and heptanoate peak intensity in mucus of individual mice. **E** Spearman correlation between colonic mucus growth rate and SCFAs, hexanoate and heptanoate peak intensity in the cecum of WSD-fed mice transplanted with human microbiota. **F** Ex vivo mucus growth rate in distal colon tissue explants with supplementation of propionate, **G** supernatant of a 24 h *B. coccoides* culture (*n* = 5) and (**H**) supernatant of a 24 h cecal community supplemented with *B. coccoides* (*n* = 6), with respective controls. **I** Quantification of acetate and propionate in supernatant of a 24 h *B. coccoides* culture (*n* = 3) and (**J**) supernatant of a 24 h cecal

community supplemented with *B. coccoides* (*n* = 3). **K** Ex vivo stimulation of distal colon tissue explants with acetate as described above (*n* = 6 mice/group). **L** Fold change in acetate and propionate peak area in supernatants of a 48 h *B. coccoides* culture incubated in GAM media supplemented with fucose (control *n* = 2; supernatant *n* = 3). **M** Ex vivo stimulation and mucus growth measurement of colonic tissue stimulated with acetate or propionate, with and without a Ffar2 chemical antagonist (*n* = 5), **N** with a chemical agonist of Ffar2 (*n* = 5) and (**O**) with supernatant of a 24–48 h *B. coccoides* culture with and without a Ffar2 chemical antagonist (*n* = 5). Statistical significance was determined by Kruskal–Wallis test and Dunn's multiple comparisons test (**F, K**) or Mann–Whitney *U* test (**C, G, H, L, M, N, O**). Normal distribution of the data was tested with the D'Agostino & Pearson test, and data are presented as mean ± SD with *p* < 0.05 (*) and *p* < 0.01 (**) considered statistically significant. All *P* values are two-sided. Source data are provided as a Source Data file.

As a fermentation product of dietary fiber, propionate has been identified as a gut microbial metabolite that induces intestinal gluconeogenesis via a gut-brain neural circuit and thereby improves host metabolism[57]. As humans and mice with metabolic impairments have been shown to display a defective colonic mucus layer[4,14] and since translocation of bacterial products or intact bacteria across the intestinal mucosal barrier has been shown to induce metabolic impairments[58–60], it is thus possible that fiber-dependent propionate production improves host metabolism, not only through intestinal gluconeogenesis but also through fortification of the colonic mucus layer.

Microbial interactions with intestinal mucus are characterized by distinct catabolic processes with opposing results[31,61]. On the one hand, and as we describe here, fiber degradation by beneficial bacteria leads to an increase in fermentation products, such as SCFAs, which can stimulate basal mucus secretion. This mucus growth is sufficient to maintain and replenish the mucus barrier while also counteracting bacterial degradation on the luminal side under homeostatic conditions. On the other hand, under conditions when dietary fiber is absent, bacterial mucus degradation increases, which eventually leads to erosion of the mucus layer[21,22,30,62,63]. When testing the fiber- and mucus-degrading capacity of our human microbiota-transplanted mice, we observed higher enzymatic activity against the dietary fiber-derived substrates β-glucopyranoside and β-xylopyranoside upon increased fiber intake from the human volunteers when the mice were fed a chow diet (see Fig. 3A). At the same time, however, in mice receiving the same microbiota transplant but instead fed a WSD, the enzymatic activity shifted toward mucin glycan degradation and especially toward the degradation of sialic acid in the human HD microbiota transplant group. Sialic acid is a terminal cap in colonic mucin glycans, and reduced mucus sialylation has been associated with congenital IBD in humans and causally linked to colitis in mice[64,65]. Maintaining sialylated mucin glycans is thus of crucial importance for the host to prevent degradation of mucin glycans. However, while individual gut bacteria have different mucin glycan degradation capabilities[21,34,66,67], comparing chow- and WSD-fed mice that received the same microbiota transplant clearly demonstrates that these enzymatic processes are influenced by the availability of dietary carbohydrate structures. Moreover, our results indicate that the increased consumption of dietary fiber in humans can direct their gut microbial community toward higher fiber degradation, thereby leaving a higher fraction of mucin glycans for mucosal barrier function.

Defective mucus function is linked to several lifestyle diseases, including obesity, metabolic diseases, and IBDs, with the diet-gut bacteria-mucus interaction being increasingly recognized as a potential contributing factor[4,6–8,14,16,68]. In our proof-of-concept study, we confirm that the microbiota from selected humans consuming this diet is not capable of inducing sufficient mucus production in mice if dietary fiber consumption is low. However, when the participants increased their daily fiber intake by approximately 14 g on average, their microbiota configuration shifted, and the beneficial microbiota-mucus interaction was restored. Notably, the supplemented dietary fiber was derived from various sources, mainly insoluble wheat and soluble oat, and was added to a broad portfolio of products, including popular convenience foods, such as pretzel breadstick and pizza[32]. The fiber-enriched food was well accepted among the participants, thereby demonstrating that changes in food composition could have beneficial outcomes for consumers and their gut bacteria without affecting consumers' food perception. While supplementation of *Blautia* to mice helped demonstrate the potential this microbe has in modulating mucus growth stimulation and to identify the underlying mechanism, externally supplemented *Blautia* would likely not establish itself in the human microbiota in the long term. The beneficial effects exerted by this microbe could therefore be promoted by increasing the baseline fiber intake rather than supplementation with *Blautia* alone. Alternatively, *Blautia* could be exploited as a new probiotic and developed to better adapt to the mucus niche for higher propionate and acetate production, similar to the improved oxygen tolerance development that has recently been reported for *Faecalibacterium prausnitzii*[69].

## Limitations of the study

In this study, we identify a mechanism by which a commensal gut bacterium can affect mucus function in mice. While our approach has several advancements, including the usage of diet-dependent human-to-mouse microbiota transplantation and measuring mucus function on viable tissue, a limitation in the study is the pooling of five human stool samples for microbiota transplantation, thereby generating an artificial community[70]. However, our main aim with this study was to identify mechanisms by which the gut microbiota contributes to mucus quality. We thus considered this reductionistic approach that limited variability to improve reproducibility. Using this strategy, we identified *Blautia* to be causally linked to improved mucus properties through the production of acetate and propionate. Pooling stool samples has also been shown to induce clinical remission and endoscopic improvement in patients with active ulcerative colitis[71,72], a disease in which mucus dysfunction is a characteristic observation[7,8]. Interestingly, *Blautia* was one of the bacterial members in the FMT donors whose abundance was associated with a significantly better response after FMT[72]. We are thus confident that our approach, despite its limitations, is appropriate to propose that increased fiber intake in humans has the potential to shape the gut bacteria into a configuration that improves mucus function. Nevertheless, such an effect will strongly depend on the individual's baseline microbiota, which, with regard to *Blautia*, may even depend on early-life breastfeeding[73]. Replicating our findings by using individual stool samples for microbiota transplantations would thus be needed to identify the best diet-microbiota combination in humans that is optimal to boost mucus function.

Another important aspect is our usage of antibiotic-treated mice instead of germ-free mice. We intentionally did not use germ-free mice, which have been shown to have a dysfunctional mucus layer[12]. However, antibiotic treatment may affect intestinal physiology[74], and little is known about the direct effect of antibiotics on mucus function. Thus, although all mice were treated the same way and the first FMTs were carried out only after a two-day washout period to allow intestinal recovery, it cannot be fully excluded that mucus function was altered by antibiotic treatment.

## Methods

### Ethical statement

All research performed in this work complies with relevant ethical regulations. Human stool samples and corresponding metabolic parameters were obtained from a fiber-intervention study that was recently conducted at the *enable* study center, ZIEL Institute for Food and Health, Technical University of Munich, Freising, Germany[32]. The study protocol was approved by the Ethical Committee of the Faculty of Medicine of the Technical University of Munich (approval no. 201/17S), and all procedures were in agreement with the Declaration of Helsinki of 1975, as revised in 2013. Written informed consent was obtained from all participants during enrollment. The protocol was registered with the German Clinical Trials Register (DRKS00013058). Animal experiments performed at Umeå University, Sweden, were approved by the local animal ethical committee (Umeå djurförsöksetiska nämnd; Dnr A14-2019).

### Human donor selection

The previously published fiber-intervention study[32] included an intervention group of 74 healthy participants (34 men and 40 women), aged between 40–65 years, with or without elevated waist circumference (>102 cm in males and >88 cm in females) who increased

their dietary fiber intake from an average of $22.5 \pm 8.0$ g/day to $36.0 \pm 9.6$ g/day through the consumption of fiber-enriched food products, mostly containing wheat and oat fibers, over a 12-week period under normal free-living conditions. Details on available food products are described in the supplementary information of the original study[32]. Stool samples were collected before and after the intervention and stored in nucleic acid-free stool collection tubes at −80 °C. For those 67 participants (33 men and 34 women) for whom all metabolic data and stool samples were available, we generated six rankings based on their improvement in waist-to-hip ratio (WHR), body mass index (BMI), free fat mass (FFM), cholesterol and triglyceride (TCA) levels and fasting glucose concentration following the fiber intervention. After combining the six rankings into one overall metabolic ranking, we quantified the compositional changes in the fecal bacterial community of the ten best metabolic responders (1 man and 9 women). To determine these changes, we first ranked the individuals based on their shifts in the Bray–Curtis dissimilarity index and weighted UniFrac distance metric and then combined the two rankings into one overall gut bacterial ranking. After combining metabolic and gut bacteria ranking, the top individuals (1 man and 4 women) were selected for microbiota transplantation. Due to limited stool, the first FMT experiment (Fig. 1) included individuals ranked 1, 2, 3, 5, and 6, while the second FMT experiment (Fig. 3) included individuals ranked 1, 2, 4, 5, and 6. The replacement did not change any result during the metabolic responses (S1E) or the male–female sex ratio.

## Mice
C57Bl/6 J mice were obtained from Charles River Laboratory (Germany) and housed at $22 \pm 1$ °C and $55\% \pm 5\%$ humidity under a 12-h day and night cycle in individually ventilated cages in a specific-pathogen-free environment. Mice were used directly for experiments (Figs. 1–3) or kept as breeders for subsequent experiments. Male and female mice were used in all experiments; details are provided in the Source data file. If not indicated otherwise, mice were fed a common chow diet (#801730, Special Diet Services, UK: 4.02 MJ/kg atwater fuel energy (AFE); 9.12% AFE from oil (20.28% saturated, 36.65% mono-unsaturated, 43.06% polyunsaturated fatty acids), 22.2% AFE from protein, 68.68% AFE from carbohydrates (total dietary fiber 14.99%; pectin 1,4%, hemicellulose 8.82%, cellulose 3.85%, lignin 1.4%, starch 42.63%, sugar 3.94%)).

## Bacterial strains
Nalidixic acid-resistant (Nal[r]) *Citrobacter rodentium* ICC169 was kindly provided by Dr. George Birchenough (Gothenburg, Sweden). Bacteria were grown aerobically overnight in Luria broth (LB) medium supplemented with 50 μg/ml nalidixic acid (Sigma–Aldrich) at 37 °C. Prior to infection, *C. rodentium* were washed twice in 1× PBS and adjusted to ~$2 \times 10^{10}$ CFUs/ml.

*Blautia coccoides* DSM935 and *Blautia wexlerae* DSM19850 were obtained from the German Collection of Microorganisms and Cell Cultures GmbH and cultured anaerobically in an anaerobic chamber (Don Whitley Scientific, UK) at 37 °C until log phase, reaching $OD_{600}$ values of approximately 2.4 and 1.2, respectively. Brain heart-infusion supplemented with 5 g/L yeast extract, 0.5 mg/L hemin, 0.2 g/L $NaHCO_3$ and 1 g/L cysteine (BHI-S) was used to culture *B. coccoides*, while mGAM (HyServe) was used for *B. wexlerae*, and aliquot stocks of each were then stored at −80 °C in either 15% glycerol or 10% DMSO. For SCFA profile analysis, *B. coccoides* was grown for 48 h in GAM supplemented with 4% rhamnose (PanReac AppliChem) or 4% fucose (Jennewein Biotechnologie GmbH) and centrifuged, and SCFAs were analyzed from the supernatant. The viability of all cultures was assessed by anaerobic plating on BHI-S or mGAM agar plates.

## Human-to-mouse fecal microbiota transplantation (FMT)
Nine-week-old male and female C57Bl/6 J mice were obtained from Charles River Laboratory (Germany). Antibiotic-mediated bacterial depletion was performed as previously described[75] with modifications. Briefly, all mice were given autoclaved drinking water supplemented with an absorbable antibiotic cocktail (ampicillin (1 mg/ml), cefoperazone (0.5 mg/ml), clindamycin (1 mg/ml)) for five days, followed by a two-day washout and then a non-absorbable antibiotic cocktail (streptomycin (1 mg/ml), neomycin (1 mg/ml), vancomycin (0.5 mg/ml)) for an additional five days, followed by a final two-day washout period.

Aliquots of 100 mg per human stool sample were suspended by gentle mixing in 1 ml of reduced PBS supplemented with 0.1% L-cysteine and 15% glycerol (Sigma–Aldrich, St. Louis, MO, USA) in an anaerobic chamber (Whitley DG250 Anaerobic Workstation). The suspension was incubated for 10 min to allow settling of larger particles, the supernatant was then collected, and 1 ml aliquots were stored at −80 °C. Five donor samples, each from before and after the intervention, were pooled into two separate suspensions, and 200 μl of these suspensions was then gavaged into the corresponding group (HD or HF) of antibiotic-pretreated mice. The gavages were repeated after 5 days using the same suspensions. Transplanted mice were further divided into two diet groups and fed either a chow diet or a WSD (#09683 Envigo Teklad diet, USA: 4.5 kcal/g; 40.6% kcal from fat (41% saturated, 52% mono-unsaturated, 7% polyunsaturated fatty acids), 40.7% from carbohydrates (sucrose 18.2% (w/v), corn starch 16.0% (w/v), maltodextrin 12.0% (w/v), cellulose 4.0% (w/v)) until sacrifice.

## Mucus growth rate measurements
The thickness and growth rate of the inner colonic mucus layer were measured as previously described[2,30]. Briefly, the distal part of the colon was gently flushed with ice-cold oxygenated (95% $O_2$ and 5% $CO_2$) Kreb's buffer (116 mM NaCl, 1.3 mM $CaCl_2$, 3.6 mM KCl, 1.4 mM $KH_2PO_4$, 23 mM $NaHCO_3$, and 1.2 mM $MgSO_4$ (pH 7.4)) to remove luminal content and unattached mucus ("outer layer"). After removal of the muscle layer, colonic tissue was mounted in a horizontal perfusion chamber system supplemented with a continuous basolateral supply of Kreb's-glucose buffer (10 mM D-glucose, 5.7 mM sodium pyruvate and 5.1 mM sodium glutamate). Black 10 μm polystyrene microspheres (Polysciences, Warrington, USA) were added on top of the mucus to visualize the surface, and Kreb's-mannitol (10 mM mannitol, 5.7 mM sodium pyruvate and 5.1 mM sodium glutamate) was added apically for hydration. The thickness of the mucus was measured using a fine glass micropipette connected to a micrometer through a stereomicroscope (Olympus) at five different locations per sample and presented as an average value (mucus thickness). To acquire the mucus growth rate, mucus thickness was measured at 0 min and after 45 min to calculate the change in thickness per minute (μm/min).

## Mucus penetrability measurements
Mucus penetrability was analyzed as previously described[2,30] with modifications. Briefly, the distal colon was prepared as described for mucus growth rate measurements. Once mounted in the perfusion chamber, the epithelium was stained with Syto 9 (Thermo Fisher, 1:500 in Kreb's-mannitol buffer), and the mucus layer was stained with wheat germ agglutinin (Thermo Fisher, 1:20 in Kreb's-mannitol buffer) and Ulex Europaeus agglutinin I (Vector Laboratories, 1:20 in Kreb's-mannitol buffer). After 15 min, the tissue was washed with Kreb's-mannitol buffer, and mucus was overlaid with 1 μm fluorescent microspheres (Thermo Fisher, 1:20 in Kreb's-mannitol buffer). Microspheres were allowed to sediment onto the mucus surface for 10 min, and the mucosal surface was then gently washed with Kreb's-mannitol buffer to remove excess microspheres. After the addition of fresh Kreb's mannitol buffer, the tissue was visualized by acquiring confocal z-stacks (5 μm steps) with a Fluor ×20/0.50 W water dipping objective on an upright ECLIPSE Ni-E Nikon A1 plus system. Two to four z-stacks were taken per mouse, and images were exported and processed with

Imaris (Version 9.9.0, Oxford Instruments) to map the epithelium, mucus, and microspheres to isosurfaces. The mucus surface was defined in each image as the location with the highest density of microspheres (mode) or alternatively by using the position Z of the mucus surface. Distances between individual microspheres within the mucus layer and the epithelial surface were extracted from 500–1000 microspheres per image, and proximity to the epithelium was analyzed using Prism 9 software (GraphPad). Additionally, the average fraction of microspheres penetrating into the areas within a 10 µm distance from the colonic epithelium was plotted for each mouse.

**Carbohydrate hydrolase activity assay**
Carbohydrate hydrolase activity was measured as previously described[21,76] with modifications. Briefly, mouse cecal content was homogenized in 0.1 M PBS (2 µl/mg, pH 6.5), and the supernatant was collected by centrifugation (12,000 × g, 10 min, 4 °C). The collected supernatant was centrifuged again (16,000 × g for 10 min at 4 °C), collected and stored at −20 °C. Protein concentration was determined using the Pierce BCA Protein Assay kit (Thermo Fisher Scientific) following the manufacturer's protocol. Briefly, cecal supernatant was diluted 1:10 in 0.1 M PBS, and absorbance was measured at 540 nm. Samples were further normalized to a protein concentration of 1 µg/µl in 0.1 M PBS in preparation for the carbohydrate hydrolase activity assay[21].

To determine the carbohydrate hydrolase activity of the cecal microbiota community, hydrolysis of the following 4-nitrophenyl (4-NP)-linked substrates was tested: 4-NP-β-glucopyranoside, 4-NP-β-xylopyranoside, 4-NP-α-galactopyranoside, 4-NP-N-acetyl-β-glucosamine and 4-NP-α-L-fucopyranoside (Carbosynth, UK). To calculate enzyme activity, a standard curve was prepared by serial two-fold dilution of 4-NP in a 0.1 M PBS solution containing pooled protein extract of all samples, reflecting the amount used in the activity assay. The assay was carried out by incubating 0.08 µg/µl of protein extract in 1.2 mM 4-NP-linked substrate in 0.1 M PBS (pH 6.5) at 37 °C for 45 min. The enzymatic reaction was stopped by adding a 1:1 volume of 0.5 M Na$_2$CO$_3$ (pH 10.5). The absorbance of free 4-NP was measured at 405 nm, and enzyme activity was calculated for each sample (µM × min$^{-1}$ × mg$^{-1}$). In addition to the 4-NP-linked substrates, a fluorogenic assay using the 4-methylumbelliferyl (4-MU)-linked substrate 4-MU-N-acetyl-α-D-neuraminic acid (sialic acid) (Carbosynth, UK) was performed. In this assay, PBS was replaced with Tris-HCl (0.1 M, pH 7.4) during all steps, and all reactions contained 0.01 mM 4-MU-linked substrate. After incubation, the reaction was stopped by adding 0.5 M Na$_2$CO$_3$ (pH 10.5) at a 5:1 ratio. The fluorescence of free 4-MU was measured at an excitation wavelength of 360 nm, and an emission wavelength of 440 nm, and the enzyme activity was calculated for each sample (µM × min$^{-1}$ × mg$^{-1}$). All measurements were performed in duplicate, and the average is presented. For CAZy normalization, the activities of the individual enzymes were added per mouse, and the relative contribution of each enzyme to the total activity was calculated.

**RNA extraction and cDNA generation**
Directly after sacrificing, small biopsies from the distal colon were collected, snap-frozen in liquid nitrogen, and stored at −80 °C. For RNA extraction, the tissue was homogenized with stainless steel beads (5 mm; Qiagen) in a TissueLyser II (Qiagen, Germany), and RNA was extracted using an RNeasy Mini kit (Qiagen). The RNA quantity and quality were determined using a Nanodrop Lite Spectrophotometer (Thermo Fisher Scientific), and 500 ng RNA/sample was reverse transcribed to cDNA with a High-Capacity cDNA Reverse Transcription Kit (Thermo Fisher Scientific) and diluted 1:7 in nuclease-free water.

**Quantitative real-time PCR analysis**
Mouse cDNA was amplified with gene-specific primers (Table 1) and a HotStarTaq Master Mix Kit (Qiagen). Amplicons were cloned and

**Table 1 | Primers used for quantitative real-time PCR analysis**

| Gene name | Primer (5'–3') | Annealing temperature (°C) |
|---|---|---|
| Muc2 | F: GAACGGGGCCATGGTCAGCA<br>R: CATAATTGGTCTGCATGCC | 60 |
| Reg3γ | F: CCTCAGGACATCTTGTGTCTGTGCTC<br>R: TCCACCTCTGTTGGGTTCATAGCC | 68 |
| DMBT1 | F: GGGGATCTCCACTGTTATCTTGA<br>R: AGAATCTGTTCCATCTGTGGGA | 60 |
| mBD3 | F: GTTGTTTGAGGAAAGGAGGC<br>R: CCACAACTGCCAATCTGACG | 56 |
| mBD4 | F: CCACTTGCAGCCTTTACCC<br>R: GCCAATCTGTCGAAAAGCGG | 63 |
| Lypd8 | F: GCCTTCACTGTCCATCTATTT<br>R: GTGACCATAGCAAGACATGCA | 60 |

inserted into a pGEM-T vector (Promega, WI) and transformed into competent DH5α *E. coli* cells. Plasmids were isolated (Qiagen Plasmid Mini Kit) and sequenced (Eurofins Genomics, Ebersberg, Germany), target copy number/ng DNA was quantified, and target-specific dilution series were prepared. To determine the copy number of specific transcripts, mouse cDNA was analyzed in a 10 µl reaction mix consisting of 1× iQ SYBR® Green Supermix (Bio-Rad, USA), 0.2 µM of each primer and 2 µl of template cDNA on a CFX Connect Real-Time System (Bio-Rad). Duplicates of samples and plasmid standards were amplified by using the following protocol: denaturation at 95 °C for 3 min, followed by 35 cycles of denaturation at 95 °C for 20 sec, gene-specific annealing temperature (Table 1) for 40 sec, and extension at 72 °C for 60 sec. A standard curve was prepared, and the transcript copy number in each sample was calculated using Bio-Rad CFX Maestro software and reported as copy number/10 ng RNA.

**Citrobacter rodentium infection**
Ten-week-old male and female C57Bl/6 J mice obtained from Charles River, Germany, were transplanted with the human microbiota as described above and switched to WSD (#09683 Envigo Teklad diet, USA). Three days after the second gavage (day 22), mice were infected with ~5 × 10$^9$ CFUs of *C. rodentium* from a PBS-washed overnight culture via oral gavage. Seven days post infection (dpi), cecal content, colonic content, liver, spleen, and mesenteric lymph nodes were collected under sterile conditions. Organs and content were weighed and homogenized in sterile PBS using a TissueLyser II (Qiagen, Germany) at 25 Hz for 50 sec. The suspension was then plated onto LB plates supplemented with 50 mg/ml nalidixic acid and incubated at 37 °C for 2 days. Bacterial colony-forming units (CFUs) were counted from fresh fecal pellets collected at days 1, 3, and 5 post infection as well as from organs and colonic content after sacrifice (7 days post infection).

**Fecal DNA extraction and 16S rDNA sequencing library generation**
Bacterial DNA was extracted from fresh stool samples by repeated bead beating as described previously[77]. Briefly, fresh fecal samples were collected, immediately frozen in liquid nitrogen, and stored at −80 °C until further use. Total DNA was extracted by using Matrix E lysis tubes (MP Biomedicals), 4% (w/v) SDS, 50 mM Tris-HCl (pH 8), 0.5 M NaCl and 50 mM EDTA and repeated bead beating using a Fast-Prep System (MPBio, CA). DNA was precipitated and purified by a QIAmp DNA mini kit (Hilden, Germany). The V4 region of the 16S rRNA gene was amplified by PCR using 515 F and 806 R primers designed for dual indexing[78] in a reaction containing 100 ng of genomic DNA, 1× Five Prime Hot Master Mix (Quantabio), 0.2 µM of each primer, 0.4 mg/ml BSA, and 5% DMSO. PCR amplification included an initial denaturation at 94 °C for 3 min, 25 cycles of denaturation at 94 °C for 45 sec, annealing at 52 °C for 60 sec and elongation at 72 °C for 90 sec, with a

final elongation step at 72 °C for 10 min. The PCR products were purified using the NucleoSpin Gel and PCR Clean-up kit (Macherey-Nagel, Germany), quantified using the Quant-iT PicoGreen dsDNA kit (Thermo Fisher Scientific) and pooled to equimolar amounts. The pooled 16 S amplicons were purified again using Mag-Bind magnetic purification beads (Omega Biotek) before denaturation of the libraries in preparation for loading into the Illumina V2 cartridge (2 × 250 bp paired-end reads) and running on an Illumina MiSeq machine (Illumina).

## 16S rDNA bacterial sequencing analysis

16S amplicon sequences from the 71 human participants, from whom microbiota data was available for both HD and HF, were reanalyzed from the published study[32], in which the V3V4 region of the 16S gene had been sequenced. Mouse samples were collected in this study as described above; three control samples were included. A total of 19,536,751 (median = 91,440) paired-end reads from mouse samples and 4,764,753 (median = 32,206) paired-end reads from human samples were used for the analyses (Supplementary Data 2). The raw sequence data were subjected to quality filtering (at least 70% of bases with a quality score of ≥25) using Fastx (http://hannonlab.cshl.edu/fastx_toolkit/index.html) and sequence quality, adapter presence was assessed using FastQC (http://www.bioinformatics.babraham.ac.uk/projects/fastqc), and adapters were removed. Analysis of the human and mouse-derived 16S amplicon sequences was performed using the QIIME2[79] (version 2022.8) pipeline and R (version 4.1.3) in R Studio (RStudio Team, version 2022.07.2). Sequences were then clustered into ASVs using the SILVA classifier (version 138)[80]. DADA2[81] was used to generate the feature table, comprising ASV-IDs and corresponding counts of detected ASVs in each sample, with a total of 9,975 unique ASVs. Low-abundance ASVs that were present in a minimum of two samples with a total ASV count of less than 10 were excluded. A total of 3116 unique ASVs remained after this criterion, and on average, ~23,887 ASVs were detected per sample (median = 13,183).

Alpha and beta diversity analyses were carried out using the abundance of the 3116 ASVs after rarefaction analysis was performed using the GUniFrac package in R. The α-diversity metrics (observed species, Shannon) and β-diversity (weighted UniFrac distance, and Bray–Curtis distance) were calculated using phyloseq[82], vegan and ape. R packages 'mia', 'miaViz', and 'vegan' were employed to compute the centered log-ratio transformation of ASV abundance data and to assess intersample distances using the Euclidean distance metric. The number of reads assigned to different taxonomic classes (phylum and genus) was calculated, and the taxonomic composition was evaluated for each sample. A total of 2715 out of 3116 (87.13%) ASVs were assigned to the genus level, while all ASVs were assigned to the phylum level. A negative control sample was included, in which only 14 unique ASVs were detected, collectively reaching a total count of 247. The ASV count per taxonomic clade per sample was normalized by dividing the ASV count by the total number of reads in the corresponding sample to generate the relative abundance. Transformation of genus and phylum abundance data into CLR-transformed values was carried out by using the taxa_transform function from the phyloseq package. A comprehensive description of the analysis, alongside codes and scripts, is available at https://doi.org/10.5281/zenodo.10848038. Statistical differences between groups were calculated by using Kruskal–Wallis or PERMANOVA and 999 permutations (beta diversity). The core microbiome at the genus level[83] was determined based on the presence of genera in 25% and 80% of samples within each group (human participants before (HD) and after (HF) the high-fiber intervention as well as mice after the human microbiota transplant). The plots were generated using ggplot2[84]. To confirm the success of the microbiota transplantation, the relative abundance of the most abundant genera was analyzed before and after FMT (Figs. 2B and Fig. 3I). The group with the highest abundance of the respective genus was set to 100%, and the remaining groups were normalized against this group.

## Targeted LC–MS metabolomics and relative intensity profiling analysis

Approximately 40 mg of cecal content (32 samples; 8 biological replicates per group) and ~5 mg of scratched mucus (10 samples; 4 + 6 biological replicates per group) were used for targeted metabolomics analysis. Cecal content and mucus samples were homogenized using the Precellys 24-bead homogenizer (1.4-mm beads; 3 cycles: 30 sec at 2.9 × $g$ with 60 sec pause at 4 °C) with 400 μl of cold extraction solvent (ACN:MeOH:MQ; 40:40:20). For SCFA profiling 100 μl of bacteria and community supernatants or respective control media were vortexed for 2 min with 400 μl of cold extraction solvent.

Subsequently, the samples were centrifuged at 18.8 × $g$ at 4 °C for 5 min. The supernatant was filtered using a Phree Phospholipid removal 96-well plate. Supernatants were evaporated under a nitrogen atmosphere, reconstituted with extraction solvent, vortexed, and chromatographed with a Thermo Vanquish UHPLC and SeQuant ZIC-pHILIC (2.1 × 100 mm, 5.0 μm particle) column (Merck). Gradient elution was carried out with a flow rate of 0.100 ml/min with 20 mM ammonium hydrogen carbonate (pH 9.4) as mobile phase A and acetonitrile as mobile phase B. Gradient elution started with 20% A for 2 min, followed by a gradual increase up to 80% A over 17 min, returning to 20% A at 17.1 min, which was maintained for up to 24 min. For SCFA profiling of bacteria and community supernatants, a Hypercarb Porous Graphitic Carbon HPLC Column, 50 × 2.1 mm 3 μm (Thermo Scientific) and a Hypercarb Guard Porous Graphitic Carbon HPLC Column, X2 10 × 4 mm 3 μm (Thermo Scientific) were used with gradient elution, starting with 0% B and held for 2.5 min, followed by a gradual increase up to 100% B until 10 min, held for 18 min, and back to 0% B at 18.1 min, where it was maintained for up to 24 min.

The Q Exactive Orbitrap quadrupole mass spectrometer was equipped with a heated electrospray ionization (HESI) source (Thermo Fischer Scientific) using polarity switching and the following settings: spray voltages of 4250 V (positive mode) and 3250 V (negative mode), sheath gas: 25 arbitrary units (AU), auxiliary gas: 15 AU, sweep gas flow 0, capillary temperature: 275 °C, S-lens RF level: 50.0. Instrument control was performed with Xcalibur 4.1.31.9 software (Thermo Fischer Scientific), and peak integration was carried out with TraceFinder 4.1 software (Thermo Fischer Scientific) using confirmed retention times for 463 metabolites standardized with a library kit MSMLS-1EA (Merck). Peak smoothening was adjusted to 7, and the sample peak threshold was set to 50,000 m/z. The data quality was monitored throughout the run using pooled QC samples prepared by pooling 5 μl of each suspended sample and interspersed throughout the run as every 10th sample. The metabolite data were checked for peak quality (poor chromatograph), relative standard deviation (20% cutoff) and carry-over (20% cutoff).

Identified metabolites were normalized according to sample weight prior to analyzing data using Metaboanalyst 5.0 (www.metaboanalyst.ca). Missing values were replaced by the limit of detection, set as 20% of the minimum positive value of each metabolite, and normalization of all samples was made by using log transformation. Metabolomic profiles of individual samples were presented as Bray–Curtis PLS-DA plots. Unsupervised hierarchical cluster analysis using Euclidean distance measurement of the 25 most altered metabolites was used to generate heatmaps. Metabolites with a fold change of 1.5 and a $p$ value of 0.05 using an unpaired nonparametric $t$ test were considered significant. Significantly altered metabolites were correlated with the measured mucus growth rate of the respective mouse by using the Pearson correlation coefficient (normally distributed data) or Spearman correlation (non-normally distributed data), and a two-tailed $t$ test with $p < 0.05$ was considered significant. Weight-

normalized raw data from the metabolomics analysis are provided in the Source Data file.

### *Blautia coccoides* supplementation in drinking water

Nine-week-old male and female C57Bl/6 J mice born and housed at the local animal facility were fed a WSD (#09683 Envigo Teklad diet, USA) and supplemented with *B. coccoides* (*n* = 6) or a respective amount of sterile growth media (*n* = 4) in drinking water. Bacteria- and media-supplemented water was exchanged every other day in the evening, and consumption was monitored. Bacterial viability in drinking water was estimated to decline from ~$3 \times 10^7$ CFUs/ml to ~$1 \times 10^3$ CFUs/ml after 48 h, leading to an average intake of ~$4.5 \times 10^7$ CFUs/day. After ~5 weeks, the mice were sacrificed, and the distal colon was collected for immediate analysis of mucus growth rate and mucus penetrability, as well as for histology. Cecal content and mucus scraped from the distal colon were collected for metabolomic analysis.

### *Blautia coccoides* and *Blautia wexlerae* supplementation through oral gavage

To compare the impact of live and dead *Blautia* on mucus function, male and female C57BL/6 mice, aged 7–10 weeks old and born and housed at the local animal facility, were fed a WSD (#09683 Envigo Teklad diet, USA) and received live or heat-killed *B. coccoides* or *B. wexlerae* via oral gavage. Heat-killing of the bacteria was performed in a 75 °C water bath for 45 min. Prior to oral administration, bacterial aliquots were centrifuged at 4000 × *g* for 5 min at 4 °C and resuspended in anaerobic reduced PBS (0.1% L-cysteine + 15% glycerol). Each mouse received 200 µl per gavage (containing ~$3 \times 10^8$ CFUs) 3 times per week across the 5-week experimental period.

### *Citrobacter rodentium* and *Blautia coccoides* coinfection

Nine-week-old male and female C57Bl/6 J mice born and housed at the local animal facility were fed a WSD (#09683 Envigo Teklad diet, USA) and supplemented with ~$6 \times 10^7$ CFUs/day of *B. coccoides* (*n* = 8) or a respective amount of sterile growth media (*n* = 8) in drinking water for 15 days. Bacteria- and media-supplemented water was exchanged daily, and consumption was monitored. On day 8 of the treatment, mice were fasted for 4 h and infected with ~$5 \times 10^9$ CFUs of *C. rodentium* by oral gavage. The *C. rodentium* load was quantified at days 1, 3, and 5 post infection by plating fecal material on LB plates supplemented with 50 mg/ml nalidixic acid. Mice were sacrificed 7 days post infection, and the cecum, liver, spleen, and colonic content were homogenized and plated on LB plates supplemented with 50 mg/ml nalidixic acid. After incubating plates at 37 °C for 2 days, the *C. rodentium* load was determined.

### Tissue histology

Distal colon pieces were fixed in Methacarn solution (60% methanol, 30% chloroform, and 10% glacial acetic acid) for at least one week. Paraffin embedding was performed using Sakura Tissue Tec VIP (USA), and 5 µm colon sections were prepared for Alcian-Blue-Periodic acid-Schiff staining. Briefly, sections were deparaffinized in xylene (VWR Chemicals) and rehydrated in gradients of ethanol (99%, 90%, and 70%) and water. Slides were run in 3% acetic acid (VWR Chemicals) and stained with Alcian blue (Sigma–Aldrich) for 20 min. Tissues were oxidized in 0.05% periodic acid (Sigma–Aldrich) before staining with Schiff's reagent (Sigma–Aldrich) for 20 min. For nuclear visualization, Mayer's hematoxylin (Sigma–Aldrich) was used, and section dehydration was performed in water, ethanol steps (70%, 90%, and 99%), and xylene. Distal colon sections were mounted with Pertex glue (Histolab), and images were captured at ×20 magnification using Pannoramic Scan P250 Flash III BL/FL (3DHistech, Hungary). The number of goblet cells per longitudinal crypt and crypt length were assessed in at least 10 crypts per sample independently by two blinded scientists.

### Ex vivo mucus growth stimulation with metabolites

Male and female C57Bl/6 J mice aged between 6 and 13 weeks, born and housed at the local animal facility, were fed a WSD (#09683 Envigo Teklad diet, USA) for 5–8 weeks, and distal colon explants were processed and prepared for mucus growth rate measurement as described above. For each mouse, the distal colon tissue was divided into two pieces, and the mucus thickness of both pieces was monitored in parallel over time, while one piece was left untreated and the other piece was treated with different concentrations of sodium propionate (Sigma–Aldrich), sodium acetate (Sigma–Aldrich), or 1 µM of the synthetic Ffar2 agonist (#371725, Calbiochem, Merck Millipore). To inhibit Ffar2, colonic tissue was preincubated on ice for 30 min in Kreb's transport buffer with 0.5 µM GLPG0974 (MedChemExpress) or DMSO control. Subsequently, the antagonist was mixed with 5 mM propionate or 50 mM acetate in mannitol buffer for apical stimulation. The mucus growth rate was calculated by comparing initial mucus thickness with mucus thickness at 45 min following growth stimulation.

### Ex vivo mucus growth stimulation with bacterial supernatant

Cecal contents from male and female C57Bl/6 J mice aged 8 weeks, fed a WSD (#09683 Envigo Teklad diet, USA) for 4–5 weeks were suspended in reduced PBS at 10 mg/ml under anaerobic conditions. The cecal microbial community was then anaerobically cultured alone or in combination with an overnight culture of *B. coccoides* (10:1 v/v ratio) in BHI-S media for 24 h. Subsequently, the WSD-derived cecal community and the community supplemented with *B. coccoides* culture were centrifuged at 5000 × *g* for 5 min, and the supernatant was sterile filtered and stored at −20 °C. In addition, *B. coccoides* was grown in BHI-S media for 24 h and centrifuged at 5000 × *g* for 5 min, and the resulting supernatant was sterile filtered and stored at −20 °C. Approximately 180 µl of the collected supernatants were used for ex vivo mucus growth rate stimulation experiments, as described above. The resulting mucus growth rate was compared to that of their respective controls. Microbial supernatants were analyzed for SCFA production as described in the metabolomics section.

### Statistics and reproducibility

Microbiota transplantation from human volunteers into mice to investigate mucus function on viable tissue is a pioneering approach, and thus no robust power calculation could be made before the experiment. However, based on experience from mouse-to-mouse microbiota transplantation[30] we calculated with group averages of 2.2 and 0.9, alpha = 0.05 and a power of 80%, resulting in 8–9 mice per group for the transplantation. For other experiments we pre-determined the number of mice based on previous experiments[30] with similar bacterial treatments that were able to detect statistically significant differences. No data were excluded from the study, except for the mucus growth rate and all related data from one mouse in the HF-Chow group, due to rupture of the colonic tissue during the mucus measurement.

Mice were randomly allocated to either group to best match age and sex. In most cases, mice were housed in groups of 3–4 mice per cage. Anaysis of mouse groups was alternated between groups during the day to exclude any circadian and/or reagent bias. Human participants for the stool sample transplantations were selected based on their metabolic parameter and shifts in microbiota compositon from the fibre intervention group. The selection/ranking of the donors is described above.

Mucus measurements of the human FMT, histological analyses and metabolomics were performed blinded. Other experiments were analyzed by the same researcher that was responsible for mouse treatment, but randomization as described above prevented any potential bias.

GraphPad Prism (version 9) was used for statistical analysis, unless stated otherwise in the methods section. Normal distribution of the

data was tested with the D'Agostino & Pearson test, and for comparisons between two groups, an unpaired *t* test was used when samples were distributed normally. For non-normally distributed samples, the Mann–Whitney *U* or Kruskal–Wallis test was used or the Wilcoxon signed-rank test for dependent samples. Correlations between mucus function and bacterial genera or metabolites were performed with Spearman correlation analysis and corrected for multiple comparisons. Details about the statistical tests are provided in the figure legends and the source data file. In all figures, each datapoint is an independent biological replicate and data are presented as mean ± SD.

### Reporting summary

Further information on research design is available in the Nature Portfolio Reporting Summary linked to this article.

## Data availability

The human data and stool samples that have been collected for a previously published study[32] are available under restricted access due to data privacy laws and can be requested through the enable cluster (skurk@tum.de). Bacterial 16S rDNA sequencing data from the human cohort have been deposited previously in the European Nucleotide Archive under the accession code PRJNA701859. Bacterial 16S rDNA data from this study have been deposited under the accession code PRJEB57076. Source data are provided with this paper.

## Code availability

A comprehensive documentation of the bacterial 16S rDNA gene sequencing analysis, including codes and scripts, can be accessed using the following link: https://doi.org/10.5281/zenodo.10848038.

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

## Acknowledgements

The authors thank Dr. George Birchenough (Gothenburg, Sweden) for valuable discussions about mucus measurements, Jonas von Hofsten (Umeå, Sweden) for support with confocal microscopy and Kertu Liis Krigul (Umeå, Sweden/Tartu, Estonia) for assistance with histology evaluation. We acknowledge the Biochemical Imaging Center at Umeå University and the National Microscopy Infrastructure, NMI(VR-RFI 2019-00217) for providing assistance in microscopy, Umeå Hypoxia Research Facility (UHRF), the Umeå Plant Science Centre (UPSC) Microscopy Facility, Umeå Centre for Comparative Biology (UCCB) and the FIMM Metabolomics/Lipidomics/Fluxomics Unit (Finland) are further acknowledged. Computations and data handling were enabled by resources provided by the Swedish National Infrastructure for Computing (SNIC) (projects 2022/23-579 and 2022/22-1059), provided by the National Academic Infrastructure for Supercomputing in Sweden (NAISS) at UPPMAX, partially funded by the Swedish Research Council through grant agreement #2018-05973. This project has been funded through the Swedish Research Council (Vetenskapsrådet, VR Starting grant (#2018-02095) to B.O.S. and V.R. grant #2021-06602 to MIMS). The authors further acknowledge financial support from the Kempe Foundation (B.O.S.), Erasmus+ (L.Z.), and NordForsk through the Nordic Research Infrastructure Hubs initiative granted to the Nordic EMBL partnership. The FIMM Metabolomics Unit was supported by HiLIFE and Biocenter Finland, and the human stool sample collection was funded by the German Ministry for Education and Research (BMBF, # 01EA1409A (T.S.)).

## Author contributions

Conceptualization: writing—original draft, funding acquisition, resources, supervision, B.O.S.; methodology: S.H., R.H.F., V.P.P.K., F.P.B., D.S., S.W., A.N., T.S., B.O.S.: investigation, S.H., R.H.F., V.P., F.P.B., D.S., S.W., L.Z., B.B., A.N., T.S., B.O.S.; writing—review & editing: all authors.

## Funding

## Competing interests

The authors declare no competing interests.
