## [Peer Review File · Nature Communications]

REVIEWERS' COMMENTS

Reviewer #1 (Remarks to the Author):

First of all, I want to thank the authors for revising the manuscript. The authors did a great job in revising the manuscript.

General statements:

- Statistical analyses have considerably improved and in my opinion, there are no concerns left.
- Phrasing of conclusions have considerably improved and I did not encounter any overstatements anymore.
- I appreciate the additional experiments with *Blautia coccooides* being gavaged and also testing another *Blautia* species.
- I also appreciate the detailed comments to my questions and concerns. Although I don't agree completely with all of the responses, this "disagreement" is just part of a natural scientific discussion and should not prevent the editor from considering this manuscript for publication.
- In my opinion, the manuscript is ready for publication.

To further clarify some things:

- The authors respond "And while we appreciate the opinion of the reviewer that experiments adding single strains should not be performed, we think that mechanistic studies require such approaches, since otherwise one would remain only with associations"

As stated in my original comment, I did not criticize "adding 1 strain" but rather "adding 1 strain into 1 specific microbiota composition", which is a considerable difference. Testing *Blautia* administration to distinct complex microbial communities would have strengthened the generalisability of the data. Given that the authors did not pay attention to the "1 specific microbiota composition" point, they also did not address this general concern in their rebuttal letter. Nevertheless, I am fine with the manuscript in the current version.

Title: I am not 100% sure about the title as the new title is a combination of in vivo and in vitro (Ffar2) work. How about: "The gut commensal *Blautia* maintains colonic mucus function under low fiber consumption through secretion of short-chain fatty acids". The mention of Ffar2 in the abstract is fine. I think the study is solid and my suggestion is that the authors could avoid something in the title that is not yet fully tested in vivo – this strategy would be only beneficial for this paper in the long run. For example, the paper will be cited mainly for the in vivo work and if someone is not interested in citing in vitro work in future, this paper might not be cited. I let the authors decide though.

Minor comments

Line 108: 16S rRNA gene sequencing

Line 193: please change to " thus give rise to the hypothesis... or raising the hypothesis... or something like this. If the hypothesis would be already supported by this line, there would be no need for further work!

Reviewer #3 (Remarks to the Author):

The reviewers have satisfactorily addressed my critiques. Out of curiosity, did the authors happen to expose their colonic tissue and mucus production model to the supernatant of cultures grown in GAM (that is, the cultures from the new experiment in 6L?) I suspect not, but would be curious about those data if they exist. (To be clear, I am not asking for such experiments to be carried out; only for such data to potentially be included if already available.)

POINT-BY-POINT RESPONSE

First, we would like to thank both reviewers again for their effort reviewing our revised manuscript.

REVIEWERS' COMMENTS

Reviewer #1 (Remarks to the Author):

First of all, I want to thank the authors for revising the manuscript. The authors did a great job in revising the manuscript.

Thank you!

General statements:

- Statistical analyses have considerably improved and in my opinion, there are no concerns left.
- Phrasing of conclusions have considerably improved and I did not encounter any overstatements anymore.
- I appreciate the additional experiments with *Blautia coccooides* being gavaged and also testing another *Blautia* species.
- I also appreciate the detailed comments to my questions and concerns. Although I don't agree completely with all of the responses, this "disagreement" is just part of a natural scientific discussion and should not prevent the editor from considering this manuscript for publication.
- In my opinion, the manuscript is ready for publication.

Thank you very much, we really appreciate this reply.

To further clarify some things:

- The authors respond "And while we appreciate the opinion of the reviewer that experiments adding single strains should not be performed, we think that mechanistic studies require such approaches, since otherwise one would remain only with associations"
- As stated in my original comment, I did not criticize "adding 1 strain" but rather "adding 1 strain into 1 specific microbiota composition", which is a considerable difference. Testing *Blautia* administration to distinct complex microbial communities would have strengthened the generalisability of the data. Given that the authors did not pay attention to the "1 specific microbiota composition" point, they also did not address this general concern in their rebuttal letter. Nevertheless, I am fine with the manuscript in the current version.

Title: I am not 100% sure about the title as the new title is a combination of in vivo and in vitro (Ffar2) work. How about: "The gut commensal *Blautia* maintains colonic mucus function under low fiber consumption through secretion of short-chain fatty acids". The mention of Ffar2 in the abstract is fine. I think the study is solid and my suggestion is that the authors could avoid something in the title that is not yet fully tested in vivo – this strategy would be only beneficial for this paper in the long run. For example, the paper will be cited mainly for the in vivo work and if someone is not interested in citing in vitro work in future, this paper might not be cited. I let the authors decide though.

Thank you for this good suggestion, we will change the title accordingly.

Minor comments

Line 108: 16S rRNA gene sequencing

Line 193: please change to "thus give rise to the hypothesis... or raising the hypothesis... or something like this. If the hypothesis would be already supported by this line, there would be no need for further work!

Thank you, we included these suggestions.

Reviewer #3 (Remarks to the Author):

The reviewers have satisfactorily addressed my critiques. Out of curiosity, did the authors happen to expose their colonic tissue and mucus production model to the supernatant of cultures grown in GAM (that is, the cultures from the new experiment in 6L?) I suspect not, but would be curious about those data if they exist. (To be clear, I am not asking for such experiments to be carried out; only for such data to potentially be included if already available.)

Thank you very much! And unfortunately, the stimulation of colonic tissue with the GAM culture supernatant has not been done (yet).